# Single-molecule analysis reveals the phosphorylation of FLS2 governs its spatiotemporal dynamics and immunity

Yaning Cui[1,2†], Hongping Qian[2†], Jinhuan Yin[2], Changwen Xu[2], Pengyun Luo[2], Xi Zhang[2], Meng Yu[3], Bodan Su[4], Xiaojuan Li[2], Jinxing Lin[1,2]*

[1]State Key Laboratory of Tree Genetics and Breeding, College of Biological Sciences and Technology, Beijing Forestry University, Beijing, China; [2]National Engineering Research Center of Tree Breeding and Ecological Restoration, College of Biological Sciences and Technology, Beijing Forestry University, Beijing, China; [3]College of Life Sciences, Hebei Agricultural University, Baoding, China; [4]Biotechnology Research Institute, Beijing, China

**\*For correspondence:**
linjx@ibcas.ac.cn

[†]These authors contributed equally to this work

**Abstract** The *Arabidopsis thaliana* FLAGELLIN-SENSITIVE2 (FLS2), a typical receptor kinase, recognizes the conserved 22 amino acid sequence in the N-terminal region of flagellin (flg22) to initiate plant defense pathways, which was intensively studied in the past decades. However, the dynamic regulation of FLS2 phosphorylation at the plasma membrane after flg22 recognition needs further elucidation. Through single-particle tracking, we demonstrated that upon flg22 treatment the phosphorylation of Ser-938 in FLS2 impacts its spatiotemporal dynamics and lifetime. Following Förster resonance energy transfer-fluorescence lifetime imaging microscopy and protein proximity indexes assays revealed that flg22 treatment increased the co-localization of GFP-tagged FLS2/FLS2$^{S938D}$ but not FLS2$^{S938A}$ with AtRem1.3-mCherry, a sterol-rich lipid marker, indicating that the phosphorylation of FLS2$^{S938}$ affects FLS2 sorting efficiency to AtRem1.3-associated nanodomains. Importantly, we found that the phosphorylation of Ser-938 enhanced flg22-induced FLS2 internalization and immune responses, demonstrating that the phosphorylation may activate flg22-triggered immunity through partitioning FLS2 into functional AtRem1.3-associated nanodomains, which fills the gap between the FLS2$^{S938}$ phosphorylation and FLS2-mediated immunity.

## eLife assessment

This potentially **important** study employs advanced imaging techniques to directly visualize molecular dynamics and of the immune receptor kinase FLS2 in specific microenvironments. The evidence supporting the ligand-induced association with remorin and the requirement of a previously reported phosphosite as presented is **solid**, although support by independent methods would be welcome. The work will be of interest to plant biologists working on cell surface receptors.

## Introduction

Plasma membrane (PM) localized receptor kinase (RK) plays a crucial role in pattern-triggered immunity (PTI), the initial defense layer in plants, features with an extracellular domain, a single transmembrane region, and a cytoplasmic kinase domain (*Liang and Zhou, 2018*). FLAGELLIN-SENSING 2 (FLS2), a star RK in PTI, senses a conserved N-terminal epitope (flg22) of the bacterial flagellin (*Hudson et al., 2024*). Upon flg22 perception, FLS2 rapidly interacts with its co-receptor brassinosteroid insensitive1-associated kinase1 (BAK1) to form an active receptor complex (*Chinchilla et al.,*

*2006*; *Chinchilla et al., 2007*; *Heese et al., 2007*), initiating phosphorylation events through activating receptor-like cytoplasmic kinases (RLCKs) such as BOTRYTIS-INDUCED KINASE 1 (BIK1) to elicit downstream immune responses (*Lu et al., 2010*; *Zhang et al., 2010*).

Protein phosphorylation, a vital post-translational modification, plays an essential role in signal transduction. Previous studies suggest that protein phosphorylation alterations affect protein dynamics and subcellular trafficking (*Kaiserli et al., 2009*; *Kontaxi et al., 2023*). In plants, phosphorylation of proteins can coalesce into membrane nanodomains, forming platforms for active protein function at the PM (*Bücherl et al., 2017*). For example, *Xue et al., 2018* demonstrated that phosphorylation of the blue light receptor phot1 accelerates the protein movement, enhancing its interaction with a sterol-rich lipid marker *At*Rem1.3-mCherry, which underscores the crucial role of protein phosphorylation in protein dynamics and membrane partitioning. Upon flg22 treatment, multiple FLS2 phosphorylation sites are activated, with the Serine-938 phosphorylation playing a pivotal role in defense activation (*Cao et al., 2013*); nevertheless, how the phosphorylation of FLS2$^{S938}$ impacts on plant immunity remains elusive.

Membrane nanodomains, which are crucial in regulating PM protein behavior, are dynamic structures enriched with sterols and sphingolipids (*Boutté and Jaillais, 2020*) and uniquely labeled by proteins such as Flotillins and Remorins within living cells (*Martinière and Zelazny, 2021*). Various stimuli are able to trigger PM proteins moving into mobile or immobile nanodomains, suggesting a connection between nanodomains and signal transduction (*Wang et al., 2015*). For example, *Xing et al., 2022* demonstrated that sterol depletion significantly impacts the dynamics of flg22-activated FER-GFP, emphasizing the role of nanodomains in the lateral mobility and dissociation of FER from the PM under flg22 treatment. However, the spatial coordination of FLS2 dynamics and signaling at the PM, and their relationships with nanodomains remain poorly understood.

To investigate whether the FLS2$^{S938}$ phosphorylation activates immune responses through nanodomains, we analyzed the diffusion and lifetime of FLS2 phospho-dead and phospho-mimic mutants at the PM before and after flg22 treatments. Our results show that flg22-induced dynamic and lifetime changes were abolished in FLS2$^{S938A}$. Using FLIM-FRET and single-molecule protein proximity index (smPPI) techniques, we found that FLS2 phosphorylation variants exhibited distinct membrane nanodomain distribution and endocytosis. Importantly, we demonstrated that the immune response of phosphor-mimic FLS2$^{S938D}$ was comparable to wild-type FLS2, while the phosphor-dead version of FLS2$^{S938A}$ weakened the immune response. Our findings pinpoint the missing piece of FLS2-mediated immune signaling in planta.

## Results and discussion

### Ser-938 phosphorylation site changed the spatiotemporal dynamics of flg22-induced FLS2 at the plasma membrane

Previous studies highlight the crucial role of membrane protein phosphorylation in fundamental cellular processes, including PM dynamics (*Vitrac et al., 2019*; *Offringa and Huang, 2013*). In vitro mass spectrometry (MS) identified multiple phosphorylation sites in FLS2. Genetic analysis further identified Ser-938 as a functionally important site for FLS2 in vivo (*Cao et al., 2013*). FLS2 Ser-938 mutations impact flg22-induced signaling, while BAK1 binding remains unaffected, thereby suggesting Ser-938 regulates other aspects of FLS2 activity (*Cao et al., 2013*). To unravel the immune response regulation mechanisms via FLS2 phosphorylation, we generated transgenic *Arabidopsis* plants expressing C-terminal GFP-fused FLS2, S938A, or S938D under the FLS2 native promoter in the *fls2* mutant background (*Figure 1—figure supplement 1*). Using VA-TIRFM with single-particle tracking (SPT) (*Figure 1A and B*, *Figure 1—source data 1*), we investigated the diffusion dynamics of FLS2 phospho-dead and phospho-mimic mutants, following previous reports (*Geng et al., 2022*). Upon flg22 treatment, FLS2/FLS2$^{S938D}$-GFP spots demonstrated extended motion trajectories, contrasting with the shorter motion tracks of FLS2$^{S938A}$-GFP spots (*Figure 1C*). The results indicate significant changes in the diffusion coefficients and motion ranges of FLS2/FLS2$^{S938D}$-GFP after flg22 treatment, whereas FLS2$^{S938A}$-GFP showed no significant differences (*Figure 1D and E* and *Figure 1—source data 2 and 3*). Similar results were obtained using Uniform Manifold Approximation and Projection (UMAP) technology (*Dorrity et al., 2020*; *Figure 1F* and *Figure 1—figure supplement 2*) and fluorescence recovery after photobleaching (FRAP) (*Greig and Bulgakova, 2021*; *Figure 1G and H*;

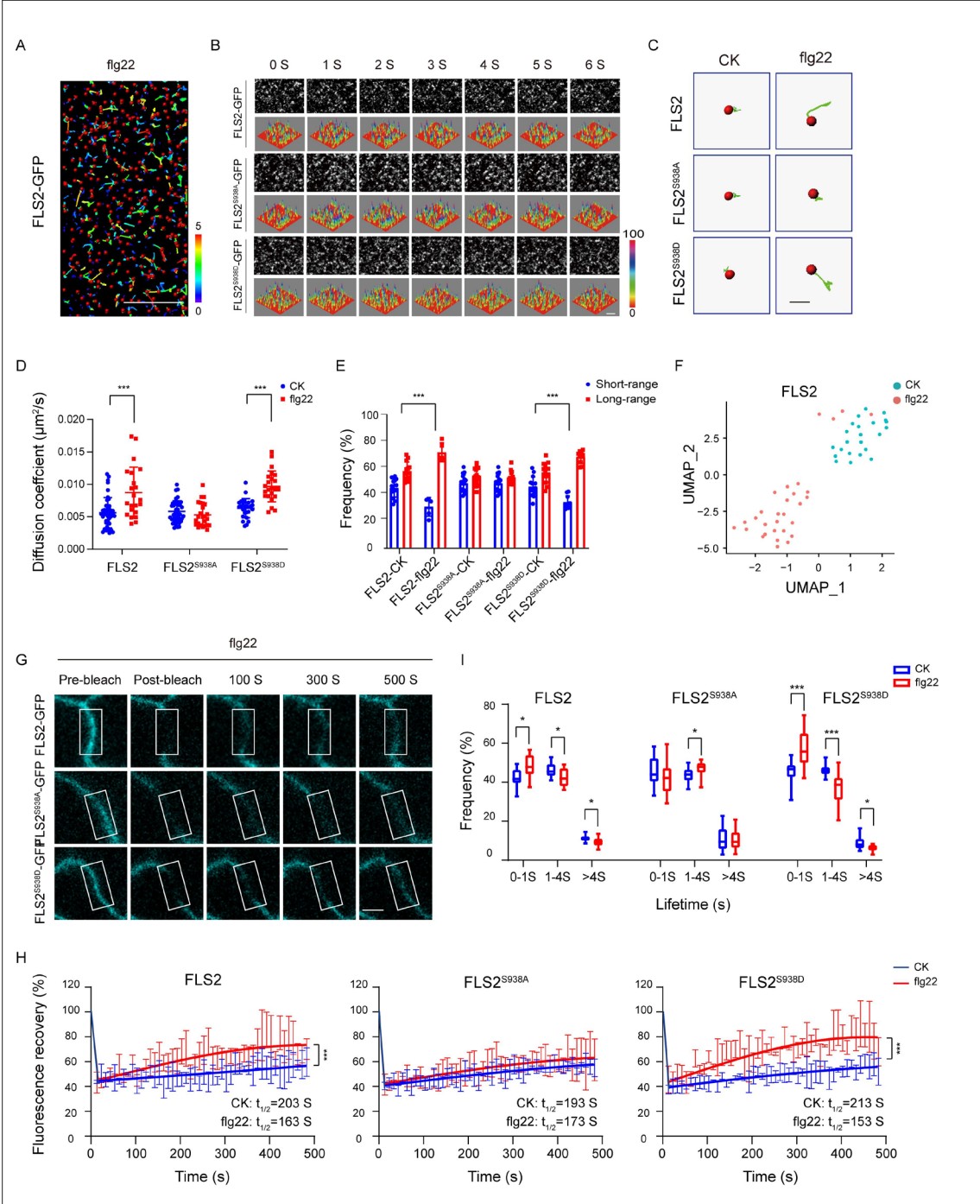

**Figure 1.** Effects of Ser-938 phosphorylation on the spatiotemporal dynamics of FLS2 at the plasma membrane. (**A**) VA-TIRFM images of a FLS2-expressing hypocotyl cell were analyzed. The 5-day-old transgenic *Arabidopsis* plant cells were observed under VA-TIRFM. The red balls indicate the positions of the identified points that appeared. Trajectories represent the track length of the identified points. Bar = 10 μm. (**B**) Time-lapse images of FLS2, FLS2$^{S938A}$, and FLS2$^{S938D}$. Bar = 2 μm. The fluorescence intensity changes among different 3D luminance plots. (**C**) The trajectories of representative individual FLS2, FLS2$^{S938A}$, and FLS2$^{S938D}$ under 30 min for 10 μM flg22 processing and control. Bar = 0.5 μm. (**D**) Diffusion coefficients of FLS2 (control, n = 42 spots; flg22, n = 22 spots), FLS2$^{S938A}$ (control, n = 44 spots; flg22, n = 25 spots), and FLS2$^{S938D}$ (control, n = 27 spots; flg22, n = 23 spots) under different environments. Statistical significance was assessed using Student's *t*-test (***p<0.001). Error bars represent the SD. (**E**) Frequency of long- and short-range motions for FLS2 (control, n = 13 spots; flg22, n = 5 spots), FLS2$^{S938A}$ (control, n = 16 spots; flg22, n = 14 spots), and FLS2$^{S938D}$ (control, n = 14 spots; flg22, n = 10 spots) under different environments. Statistical significance was assessed using Student's *t*-test (***p<0.001). Error bars represent the SD. (**F**) Uniform Manifold Approximation and Projection (UMAP) visualization of FLS2 samples under different conditions (control, n = 25 spots; flg22, n = 22 spots). Dots represent the individual images and are colored according to the reaction conditions. (**G**) The representative FRAP time course

*Figure 1 continued on next page*

*Figure 1 continued*

of FLS2-GFP, FLS2$^{S938A}$, and FLS2$^{S938D}$ under flg22 treatments. White squares indicate bleached regions. Bar = 5 µm. (**H**) Fluorescence recovery curves of the photobleached areas with or without the flg22 treatment. Three biological replicates were performed. Each experiment was repeated thrice independently (n = 3 images). Statistical significance was assessed using Student's *t*-test (***p<0.001). Error bars represent the SD. (**I**) Lifetime were analyzed for FLS2 (control, n = 13 spots; flg22, n = 9 spots), FLS2$^{S938A}$ (control, n = 28 spots; flg22, n = 21 spots), and FLS2$^{S938D}$ (control, n = 13 spots; flg22, n = 15 spots) under the control and flg22 treatments. The data points were collected from a TIRFM time series using an exposure time of 100 ms to capture a total duration of 20 s. Statistical significance was assessed using Student's *t*-test (*p<0.05, ***p<0.001). Error bars represent the SD.

The online version of this article includes the following source data and figure supplement(s) for figure 1:

**Source data 1.** The original VA-TIRFM image and single-particle tracking of FLS2-expressing hypocotyl cells are shown in *Figure 1A*.

**Source data 2.** The list of the diffusion coefficients of FLS2/FLS2$^{S938A}$/FLS2$^{S938D}$-GFP under different treatments is shown in *Figure 1D*.

**Source data 3.** The list of the motion range of FLS2/FLS2$^{S938A}$/FLS2$^{S938D}$-GFP under different treatments is shown in *Figure 1E*.

**Source data 4.** The list of the lifetime of FLS2/FLS2$^{S938A}$/FLS2$^{S938D}$-GFP under different treatments is shown in *Figure 1I*.

**Figure supplement 1.** The subcellular localization of FLS2-GFP, FLS2$^{S938A}$-GFP, and FLS2$^{S938D}$-GFP was determined by confocal imaging.

**Figure supplement 1—source data 1.** The original confocal images of FLS2-GFP, FLS2$^{S938A}$-GFP, and FLS2$^{S938D}$-GFP.

**Figure supplement 2.** Uniform Manifold Approximation and Projection (UMAP) visualization of FLS2$^{S938A}$ (control, n = 27 spots; flg22, n = 30 spots) and FLS2$^{S938D}$ (control, n = 11 spots; flg22, n = 14 spots) samples in the different conditions.

**Figure supplement 3.** The representative fluorescence recovery after photobleaching (FRAP) time course of FLS2-GFP, FLS2$^{S938A}$, and FLS2$^{S938D}$ under control treatments.

**Figure supplement 4.** Images of FLS2-GFP, FLS2$^{S938A}$-GFP, and FLS2$^{S938D}$-GFP with flg22 treatment were collected by VA-TIRFM.

**Figure supplement 5.** Representative kymographs showing individual FLS2-GFP, FLS2$^{S938A}$-GFP, and FLS2$^{S938D}$-GFP dwell times under the control and flg22 treatment.

**Figure supplement 6.** The single-molecule trajectories of FLS2-GFP, FLS2$^{S938A}$-GFP, and FLS2$^{S938D}$-GFP analyzed by Imaris could be faithfully tracked under control and flg22 treatments.

*Figure 1—figure supplement 3*), supporting the essential role of the S938 phosphorylation site in flg22-induced lateral diffusion of FLS2 at the PM.

Using VA-TIRFM, we analyzed FLS2 particle lifetime across various phosphorylation states. The results revealed that flg22 treatment significantly reduced the fluorescence trajectory of FLS2 molecules compared to the control (*Figure 1—figure supplement 4*), indicating that Ser-938 phosphorylation influences flg22-induced lifetime of FLS2 at the PM. Subsequently, real-time dynamic analysis through the Kymograph technique provided spatiotemporal information via frame-by-frame tracking (*Zhou et al., 2020*; *Su et al., 2023*). Compared to FLS2-GFP and FLS2$^{S938D}$-GFP, FLS2$^{S938A}$-GFP showed nearly linear fluctuations in fluorescence intensity under flg22 conditions, and the duration of fluorescence retention was essentially unchanged (*Figure 1—figure supplement 5*). To validate this, we used an exposure time of 100 ms to capture a time series with a total duration of 20 s. Therefore, we divided the times into three segments: 0–1 s, 1–4 s, and >4 s (*Bücherl et al., 2017*). Numerous FLS2 molecules exhibited a short-lived lifetime, which could be attributed to fluorescence bleaching or potentially reflect the occurrence of abortive endocytic events (*Bertot et al., 2018*). Additionally, we focused on lifetimes exceeding 4 s. The results showed after flg22 treatment the lifetime of FLS2$^{S938D}$-GFP greater than 4S significantly decreased, resembling that of FLS2-GFP. While FLS2$^{S938A}$-GFP plants showed a minor decrease in lifetime upon flg22 treatment, this change was insignificant (*Figure 1I*, *Figure 1—source data 4*, *Figure 1—figure supplement 6*). This aligns with previous findings that indicated that NRT1.1 phosphorylation affects dynamics and lifetime (*Zhang et al., 2019*). Therefore, these results underscore the impact of Ser-938 phosphorylation on the spatiotemporal dynamics of flg22-induced FLS2.

## Ser-938 phosphorylation enhances recruitment of FLS2/BAK1 heterodimerization into *At*Rem1.3-associated nanodomains

Proteins rarely act independently; they typically form multimers to enhance downstream signaling (*Li et al., 2022*). Flg22 treatment induces FLS2 and BAK1 to heterodimerize at the PM, signifying flg22 as a ligand promoting this heterodimerization (*Orosa et al., 2018*). To investigate the impact of Ser-938 phosphorylation on FLS2/BAK1 heterodimerization, we employed Tesseler technology, analyzing FLS2 and BAK1 co-localization in *Nicotiana benthamiana* epidermal cells. Partial co-localization of

FLS2-GFP, FLS2$^{S938A}$-GFP, and FLS2$^{S938D}$-GFP with BAK1-mCherry was observed after treatment (*Figure 2—figure supplement 1*), suggesting that heterodimerization formation is independent of Ser-938 phosphorylation. To further verify if FLS2/BAK1 heterodimerization requires Ser-938 phosphorylation, we analyzed the donor fluorescence lifetime (t) and corrected fluorescence resonance energy transfer (FRET) efficiency (IPS value). A significant decrease in GFP lifetime (t) was observed in leaves expressing FLS2/FLS2$^{S938D}$/FLS2$^{S938A}$ and BAK1-mCherry upon flg22 application (*Figure 2A*, *Figure 2—source data 1*). The FRET efficiency (IPS) between FLS2, FLS2$^{S938D}$, and FLS2$^{S938A}$ with BAK1-mCherry significantly increased after flg22 treatment, suggesting that the FLS2 Ser-938 phosphorylation site is not essential for flg22-induced heterodimerization. To validate this observation, we quantified FLS2/FLS2$^{S938D}$/FLS2$^{S938A}$ and BAK1-mCherry co-localization using Pearson correlation coefficients. A marked increase in the coefficient after flg22 treatment for FLS2/FLS2$^{S938D}$/FLS2$^{S938A}$ and BAK1-mCherry (*Figure 2B*, *Figure 2—source data 2*, *Figure 2—figure supplement 2*) indicated flg22-induced interaction between FLS2-GFP and BAK1-mCherry, irrespective of FLS2 Ser-938 phosphorylation. Additionally, we applied protein proximity indexes (PPIs) to estimate the degree of co-localization between FLS2/FLS2$^{S938D}$/FLS2$^{S938A}$ and BAK1-mCherry. After flg22 treatment, mean PPIs of FLS2, FLS2$^{S938D}$, and FLS2$^{S938A}$ with BAK1 increased (*Figure 2C*, *Figure 2—source data 3*, *Figure 2—figure supplement 3*), thereby further supporting our findings. This aligns with the previous finding that flg22 acts as a molecular glue for FLS2 and BAK1 ectodomains (*Sun et al., 2013*), confirming the independence of FLS2/BAK1 heterodimerization from phosphorylation, with these events occurring sequentially.

Membrane nanodomains serve as pivotal platforms for protein regulation, impacting cellular signaling dynamics (*Martinière and Zelazny, 2021*). Studies reveal that specific PM proteins, responsive to developmental cues and environmental stimuli, exhibit dynamic movements within and outside these nanodomains (*Lee et al., 2019*). For instance, treatment with the secreted peptides RAPID ALKALINIZATION FACTOR (RALF1 and RALF23) enhances the presence of FERONIA (FER) in membrane nanodomains (*Gronnier et al., 2022*). Conversely, flg22-activated BSK1 translocates from membrane nanodomains to non-membrane nanodomains (*Su et al., 2021*). In a previous investigation, we demonstrated that flg22 induces FLS2 translocation from AtFlot1-negative to AtFlot1-positive nanodomains in the plasma membrane, implying a connection between FLS2 phosphorylation and membrane nanodomain distribution (*Cui et al., 2018*). To validate this, we assessed the association of FLS2/FLS2$^{S938D}$/FLS2$^{S938A}$ with membrane nanodomains, using *At*Rem1.3-associated nanodomains as representatives (*Lv et al., 2017*; *Huang et al., 2019*). Upon observation using dual-color TIRM-SIM, partial co-localization was detected between FLS2-GFP, FLS2$^{S938D}$-GFP, FLS2$^{S938A}$-GFP, and *At*Rem1.3-mCherry foci on cell surfaces (*Figure 2D*). SPT was employed to quantify the co-localization ratio, revealing a small overlap in the cross-correlation signal for FLS2$^{S938A}$-GFP and *At*Rem1.3-mCherry after flg22 treatment, similar to untreated cells. In contrast, flg22-treated seedlings displayed higher cross-correlation signals for FLS2-GFP/FLS2$^{S938D}$-GFP and *At*Rem1.3-mCherry compared to untreated seedlings (*Figure 2E and F*), suggesting that phosphorylation increased the movement of FLS2 protein to membrane nanodomains. To further verify this result, FRET-FLIM was applied to examine the correlation between FLS2/FLS2$^{S938D}$/FLS2$^{S938A}$-GFP and *At*Rem1.3-mCherry. The results showed that, after flg22 treatment, plants co-expressing FLS2$^{S938D}$-GFP with *At*Rem1.3-mCherry had a strongly reduced average GFP fluorescence lifetime compared to control seedlings, which was similar to that of FLS2-GFP seedlings (*Figure 2G and H*, *Figure 2—source data 4 and 5*). Meanwhile, the flg22 treatment increased the IPS for FLS2 and FLS2$^{S938D}$ with *At*Rem1.3-mCherry. No significant difference was observed in the fluorescence lifetime and IPS of FLS2$^{S938A}$-GFP co-expressed with AtRem1.3-mCherry, with or without flg22 treatment (*Figure 2G and H*, *Figure 2—source data 4 and 5*), suggesting that Ser-938 phosphorylation may influence FLS2 partitioning into PM nanodomains. To further investigate the role of phosphorylation in FLS2 distribution into nanodomains, we used Pearson correlation coefficients to quantify the co-localization between FLS2/FLS2$^{S938D}$/FLS2$^{S938A}$-GFP and *At*Rem1.3-mCherry. It was found that the mean co-localization values between FLS2-GFP/FLS2$^{S938D}$-GFP and *At*Rem1.3-mCherry under flg22 treatment were significantly higher compared to the control group (*Figure 2I*, *Figure 2—source data 6*). Notably, FLS2$^{S938A}$-GFP and *At*Rem1.3-mCherry demonstrated similar co-localization values under both control group and flg22 conditions (*Figure 2I*, *Figure 2—source data 6*), indicating that the phosphorylated form of FLS2 may modify the distribution of PM nanodomains. These findings suggest that the phosphorylation state of FLS2 at Ser-938 influences its

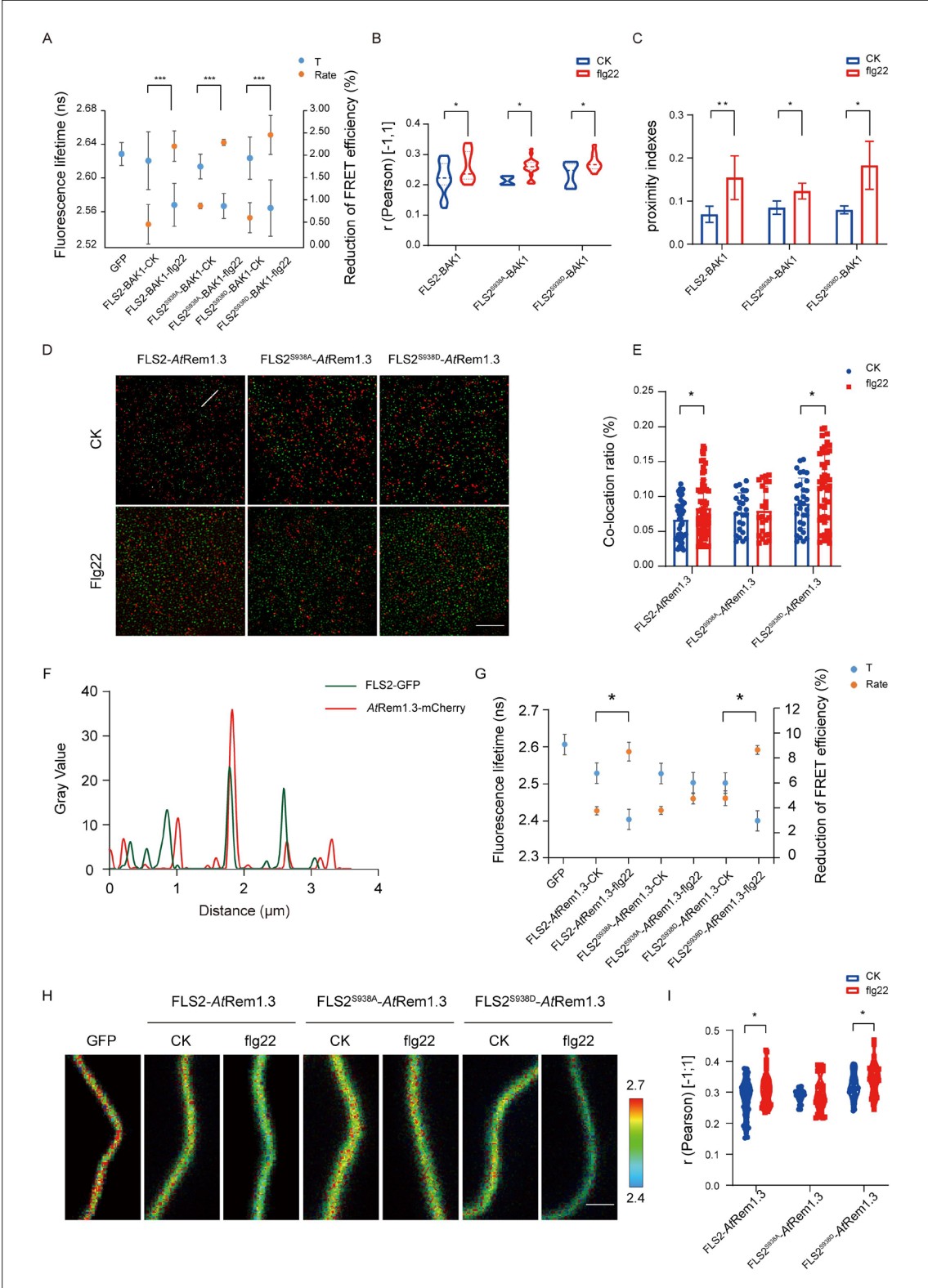

**Figure 2.** Different Ser-938 phosphorylation states of FLS2 affect its partitioning into *At*Rem1.3-associated nanodomains. (**A**) FLIM-FRET was used to detect the co-expression of FLS2/FLS2^S938A^/FLS2^S938D^-GFP and BAK1-mCherry in the *N. benthamiana* epidermal cells stimulated by 1/2 MS or flg22 (10 μM) for 30 min. Average fluorescence lifetime (t) and the FRET efficiency were analyzed for FLS2 (control, n = 21 images; flg22, n = 14 images), FLS2^S938A^ (control, n = 18 images; flg22, n = 12 images), or FLS2^S938D^ (control, n = 24 images; flg22, n = 14 images) and BAK1. The fluorescence mean lifetime (t) of FLS2-GFP (n = 7 images). Statistical significance was assessed using Student's *t*-test (***p<0.001). Error bars represent the SD. (**B**) Pearson

*Figure 2 continued on next page*

*Figure 2 continued*

correlation coefficient values of co-localization between FLS2 (control, n = 21 images; flg22, n = 11 images), FLS2$^{S938A}$ (control, n = 4 images; flg22, n = 20 images), or FLS2$^{S938D}$ (control, n = 8 images; flg22, n = 10 images) and BAK1 upon stimulation with control or flg22. Statistical significance was assessed using Student's *t*-test (*p<0.05). Error bars represent the SD. (**C**) Mean protein proximity indexes showing FLS2 (control, n = 6 images; flg22, n = 5 images), FLS2$^{S938A}$ (control, n = 4 images; flg22, n = 3 images), or FLS2$^{S938D}$ (control, n = 4 images; flg22, n = 3 images) and BAK1 degree of proximity under different conditions. Statistical significance was assessed using Student's *t*-test (*p<0.05, **p<0.01). Error bars represent the SD. (**D**) TIRF-SIM images of the *Arabidopsis* leaf epidermal cells co-expressing FLS2/FLS2$^{S938A}$/FLS2$^{S938D}$-GFP and *At*Rem1.3-mCherry under different environments. The distribution of FLS2 and *At*Rem1.3 signals along the white line in the merged image. Bar = 5 µm. (**E**) The histogram shows the co-localization ratio of FLS2-GFP (control, n = 46 images; flg22, n = 59 images), FLS2$^{S938A}$-GFP (control, n = 24 images; flg22, n = 36 images), or FLS2$^{S938D}$-GFP (control, n = 30 images; flg22, n = 54 images) and *At*Rem1.3-mCherry. The sizes of the ROIs used for statistical analysis are 13.38 µm and 13.38 µm. Statistical significance was assessed using Student's *t*-test (*p<0.05). Error bars represent the SD. (**F**) FLS2/FLS2$^{S938A}$/FLS2$^{S938D}$-GFP and *At*Rem1.3-mCherry fluorescence signals as shown in (**D**). (**G**) The fluorescence mean lifetime (t) and the corrected fluorescence resonance efficiency (Rate) of FLS2 (control, n = 8 images; flg22, n = 13 images), FLS2$^{S938A}$ (control, n = 9 images; flg22, n = 11 images), or FLS2$^{S938D}$ (control, n = 10 images; flg22, n = 11 images) with co-expressed *At*Rem1.3. The fluorescence mean lifetime (t) of FLS2-GFP (n = 9 images). Statistical significance was assessed using Student's *t*-test (***p<0.001). Error bars represent the SD. (**H**) Intensity and lifetime maps of the *Arabidopsis* leaf epidermal cells co-expressing FLS2/FLS2$^{S938A}$/FLS2$^{S938D}$-GFP and *At*Rem1.3-mCherry as measured by FLIM-FRET. Bar = 2 µm. (**I**) Quantification of co-localization between FLS2, FLS2$^{S938A}$ or FLS2$^{S938D}$ and *AtRem1.3* with and without stimulation with ligands. Pearson correlation coefficient values (r) were obtained between FLS2 (control, n = 62 images; flg22, n = 53 images)/FLS2$^{S938A}$ (control, n = 26 images; flg22, n = 40 images)/FLS2$^{S938D}$-GFP (control, n = 28 images; flg22, n = 38 images) and *At*Rem1.3-mCherry. Statistical significance was assessed using Student's *t*-test (*p<0.05). Error bars represent the SD.

The online version of this article includes the following source data and figure supplement(s) for figure 2:

**Source data 1.** The list of the fluorescence mean lifetimes (t) of FLS2/FLS2$^{S938A}$/FLS2$^{S938D}$-GFP and BAK1-mCherry under different treatments is shown in *Figure 2A*.

**Source data 2.** The list of the Pearson correlation coefficient of FLS2/FLS2$^{S938A}$/FLS2$^{S938D}$-GFP and BAK1-mCherry under different treatments is shown in *Figure 2B*.

**Source data 3.** The list of the mean protein proximity indexes of FLS2/FLS2$^{S938A}$/FLS2$^{S938D}$-GFP and BAK1-mCherry under different treatments is shown in *Figure 2C*.

**Source data 4.** The original intensity and lifetime maps of FLS2/FLS2$^{S938A}$/FLS2$^{S938D}$-GFP and *At*Rem1.3-mCherry under different treatments are shown in *Figure 2H*.

**Source data 5.** The list of the fluorescence mean lifetimes (t) of FLS2/FLS2$^{S938A}$/FLS2$^{S938D}$-GFP and *At*Rem1.3-mCherry under different treatments is shown in *Figure 2G*.

**Source data 6.** The list of the Pearson correlation coefficient of FLS2/FLS2$^{S938A}$/FLS2$^{S938D}$-GFP and *At*Rem1.3-mCherry under different treatments is shown in *Figure 2I*.

**Figure supplement 1.** SR-Tesseler analysis shows the distribution and co-localization of all spots on FLS2-GFP, FLS2$^{S938A}$-GFP and FLS2$^{S938D}$-GFP and BAK1 under control or flg22 treatment.

**Figure supplement 1—source data 1.** The list of the coordinates of FLS2/FLS2$^{S938A}$/FLS2$^{S938D}$-GFP and BAK1-mCherry under different treatments.

**Figure supplement 2.** Images show co-localization of FLS2-GFP, FLS2$^{S938A}$-GFP, or FLS2$^{S938D}$-GFP with BAK1 in *N. benthamiana* leaves cells under control or flg22 treatment.

**Figure supplement 3.** 3D plot of FLS2-GFP, FLS2$^{S938A}$-GFP, or FLS2$^{S938D}$-GFP and BAK1 cross-correlation versus pixel shift under control or flg22 treatment.

aggregation into *At*Rem1.3-associated nanodomains, providing an efficient mechanism for triggering the immune response.

## Ser-938 phosphorylation maintains FLS2 protein homeostasis via flg22-induced endocytosis

The PM protein endocytosis is crucial for regulating intercellular signal transduction in response to environmental stimuli. Notably, Thr 867 mutation, a potential phosphorylation site on FLS2, results in impaired flg22-induced endocytosis, underscoring the significance of phosphorylation in FLS2 endocytosis (*Robatzek et al., 2006*). As shown in *Figure 1I*, both FLS2 and FLS2 phospho-mimetic mutants showed a reduced lifetime under flg22 treatment, implying a probable connection between FLS2 lifetime and endocytosis. We further found that FLS2/FLS2$^{S938D}$/FLS2$^{S938A}$-GFP accumulated in brefeldin A (BFA) compartments labeled with FM4-64, indicating that various Ser-938 phosphorylation states of FLS2 undergo BFA-dependent constitutive endocytosis (*Figure 3A*, *Figure 3—source data 1*, *Figure 3—figure supplement 1*).

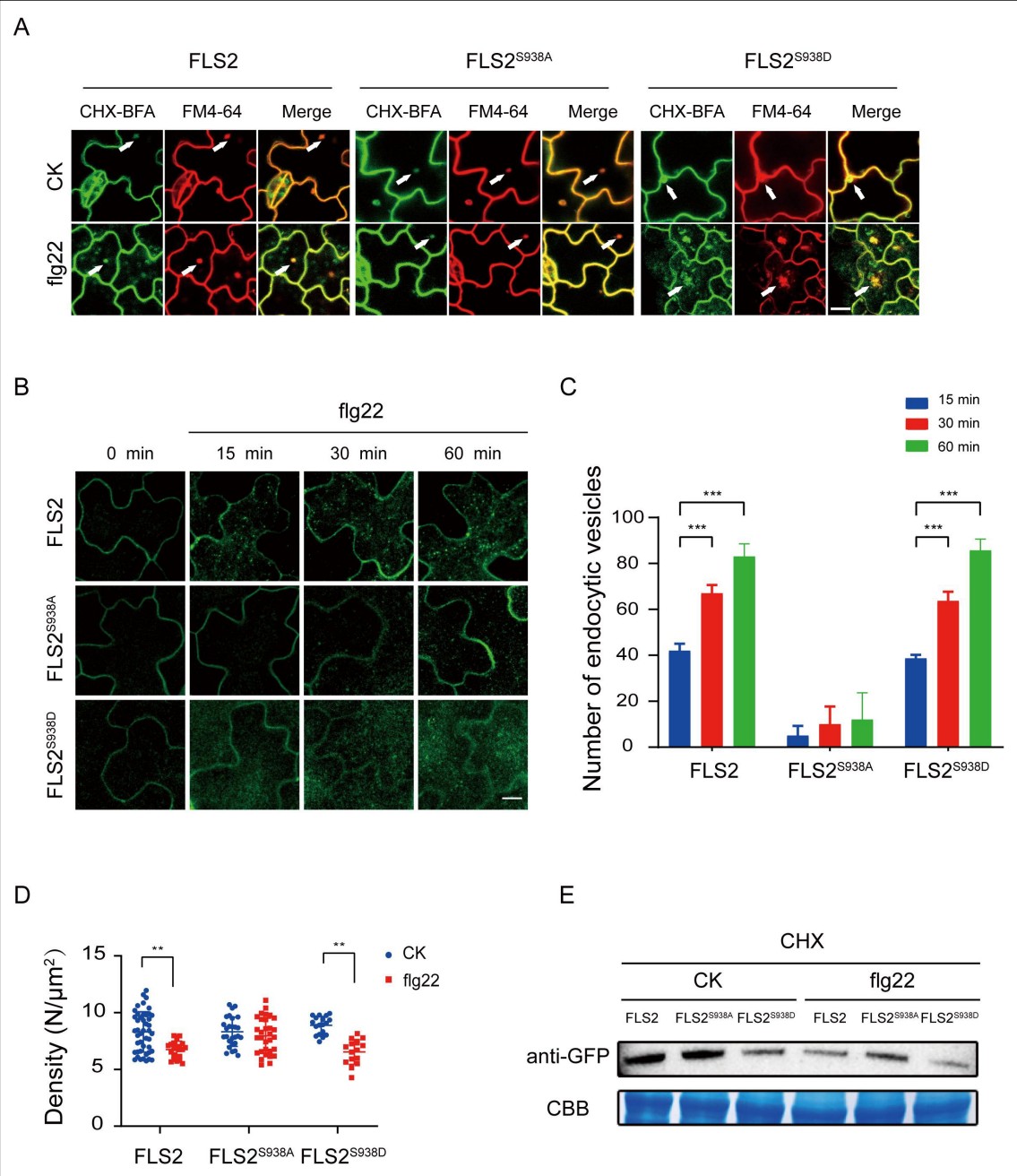

**Figure 3.** Ser-938 phosphorylation site affects flg22-induced endocytosis. (**A**) Confocal images of FLS2/FLS2$^{S938A}$/FLS2$^{S938D}$-GFP in *Arabidopsis thaliana* leaf epidermal cells. Firstly, experiments were performed after pretreatment with CHX (50 μM) for 30 min. Subsequently, to observe subcellular localization, FLS2/FLS2$^{S938A}$/FLS2$^{S938D}$-GFP were treated with 60 min of BFA (50 μM) and then exposed or not to flg22 (10 μM). Finally, the transgenic seedlings were stained with FM4-64 (5 μM, 30 min). White arrows indicate BFA bodies. Bar = 3 μm. (**B**) Images of FLS2/FLS2$^{S938A}$/FLS2$^{S938D}$-GFP in *Arabidopsis thaliana* leaf epidermal cells treated with 10 μM flg22 for 15, 30, and 60 min. Bar = 3 μm. (**C**) Analysis of FLS2 (15 min, n = 4 images; 30 min, n = 4 images; 60 min, n = 4 images), FLS2$^{S938A}$ (15 min, n = 3 images; 30 min, n = 3 images; 60 min, n = 3 images), and FLS2$^{S938D}$ (15 min, n = 3 images; 30 min, n = 3 images; 60 min, n = 3 images) endocytic vesicle numbers in cells treated with 10 μM flg22 treatment over time. Statistical significance was assessed using Student's *t*-test (\*\*\*p<0.001). Error bars represent the SD. (**D**) The signal density of FLS2 (control, n = 45 images; flg22, n = 24 images), FLS2$^{S938A}$ (control, n = 27 images; flg22, n = 31 images), and FLS2$^{S938D}$ (control, n = 18 images; flg22, n = 15 images) in cells after control or 10 μM flg22 treatments for 30 min as measured by fluorescence correlation spectroscopy (FCS). Statistical significance was assessed using Student's *t*-test (\*\*p<0.01). Error bars represent the SD. (**E**) Immunoblot analysis of FLS2 protein in 10-day-old transgenic *Arabidopsis* plant upon stimulation with or without 10 μM flg22. CBB, a loading control dyed with Coomassie Brilliant Blue.

The online version of this article includes the following source data and figure supplement(s) for figure 3:

*Figure 3 continued on next page*

*Figure 3 continued*

**Source data 1.** The original confocal images of FLS2/FLS2^S938A/FLS2^S938D-GFP under different treatments are shown in *Figure 3A*.

**Source data 2.** The original confocal images of FLS2/FLS2^S938A/FLS2^S938D-GFP in *Arabidopsis thaliana* leaf epidermal cells treated with 10 µM flg22 for 15, 30, and 60 min are shown in *Figure 3B*.

**Source data 3.** The list of the endocytic vesicle numbers of FLS2/FLS2^S938A/FLS2^S938D-GFP under different treatments is shown in *Figure 3C*.

**Source data 4.** The list of the signal density of FLS2/FLS2^S938A/FLS2^S938D-GFP under different treatments is shown in *Figure 3D*.

**Source data 5.** The original file for the western blot analysis in *Figure 3E*.

**Source data 6.** The original file for the western blot analysis with highlighted bands and sample labels is shown in *Figure 3E*.

**Figure supplement 1.** Distribution and co-localization with FM4-64 of FLS2-GFP, FLS2^S938A-GFP, and FLS2^S938D-GFP in leaves epidermal cells treated with or without 10 µM flg22 for 30 min after 30 min pretreatment with CHX.

**Figure supplement 1—source data 1.** The list of the number, diameter, and fluorescence intensity of FLS2/FLS2^S938A/FLS2^S938D-GFP BFA bodies under different treatments.

**Figure supplement 2.** Confocal images of FLS-GFP, FLS2^S938A-GFP, and FLS2^S938D-GFP in leaves epidermal cells treated with or without 10 µM flg22 after 30 min pretreatment with BFA.

**Figure supplement 3.** Effects of Ser-938 phosphorylation on the endocytosis of FLS2.

**Figure supplement 4.** The fluorescence intensity of FLS2-GFP (0 min, n = 9 images; 15 min, n = 5 images; 30 min, n = 7 images; 60 min, n = 9 images), FLS2^S938A-GFP (0 min, n = 5 images; 15 min, n = 5 images; 30 min, n = 5 images; 60 min, n = 5 images), and FLS2^S938D-GFP (0 min, n = 10 images; 15 min, n = 6 images; 30 min, n = 7 images; 60 min, n = 8 images) in the cytoplasm relative to the sum of the cytoplasm and PM in leaves epidermal cells treated with flg22 treatment over time; Three biological replicates were performed.

**Figure supplement 4—source data 1.** The list of the fluorescence intensity of FLS2-GFP, FLS2^S938A-GFP, and FLS2^S938D-GFP in the cytoplasm relative to the sum of the cytoplasm and PM in leaves epidermal cells under different treatments.

**Figure supplement 5.** The confocal images of FLS2-GFP signal density in *Arabidopsis* leaf epidermal cells upon ligand stimulation were analyzed using fluorescence correlation spectroscopy.

Next, we investigated the impact of Ser-938 on flg22-induced FLS2 internalization. Under co-treatment with BFA and flg22, FLS2/FLS2^S938D produced strong signals in both the BFA compartments. Interestingly, we found that FLS2^S938A-GFP only produced BFA compartments, and most remained at the PM region with few intracellular puncta upon stimulation with BFA and flg22 (*Figure 3—figure supplements 2 and 3*). We further tracked FLS2 endocytosis and quantified vesicle numbers over time; the results showed that both FLS2 and FLS2^S938D vesicles appeared 15 min after-flg22 treatment, significantly increasing thereafter. Notably, only a few vesicles were detected in FLS2^S938A-GFP, indicating Ser-938 phosphorylation of FLS2 impact on flg22-induced FLS2 endocytosis (*Figure 3B and C*, *Figure 3—source data 2*, *Figure 3—source data 3*, *Figure 3—figure supplement 4*). Results revealed increased endocytic vesicles for FLS2 during flg22 treatment, aligning with previous studies (*Leslie and Heese, 2017*; *Loiseau and Robatzek, 2017*). Additionally, fluorescence correlation spectroscopy (FCS) (*Chen et al., 2009*) monitored molecular density of FLS2 changes on the PM before and after flg22 treatment (*Figure 3—figure supplement 5*). *Figure 3D* shows that both FLS2-GFP and FLS2^S938D-GFP densities significantly decreased after flg22 treatment, while FLS2^S938A-GFP exhibited minimal changes, indicating Ser-938 phosphorylation affects FLS2 internalization (*Figure 3D*, *Figure 3—source data 4*). Western blotting confirmed that Ser-938 phosphorylation influences FLS2 degradation after flg22 treatment (*Figure 3E*, *Figure 3—source data 5 and 6*), consistent with single-molecule analysis findings. Therefore, our results strongly support the notion that Ser-938 phosphorylation expedited FLS2 internalization, potentially regulating its immune response capacity.

## Ser-938 phosphorylation affects FLS2-mediated responses

PTI plays a pivotal role in host defense against pathogenic infections (*Macho and Zipfel, 2014*; *Ding et al., 2022*; *Ngou et al., 2022*). Previous studies demonstrated that the perception of flg22 by FLS2 initiates a complex signaling network with multiple parallel branches, including calcium burst, mitogen-activated protein kinases (MAPKs) activation, callose deposition, and seedling growth inhibition (*Chi et al., 2021*; *Zipfel et al., 2004*; *Li et al., 2016*; *Sanguankiattichai et al., 2022*). Our focus was to investigate the significance of Ser-938 phosphorylation in flg22-induced plant immunity. Our results illustrate diverse immune responses in FLS2 and FLS2^S938D plants following flg22 treatment (*Figure 4A–F*, *Figure 4—source data 1–3*, *Figure 4—figure supplements 1–3*). These responses

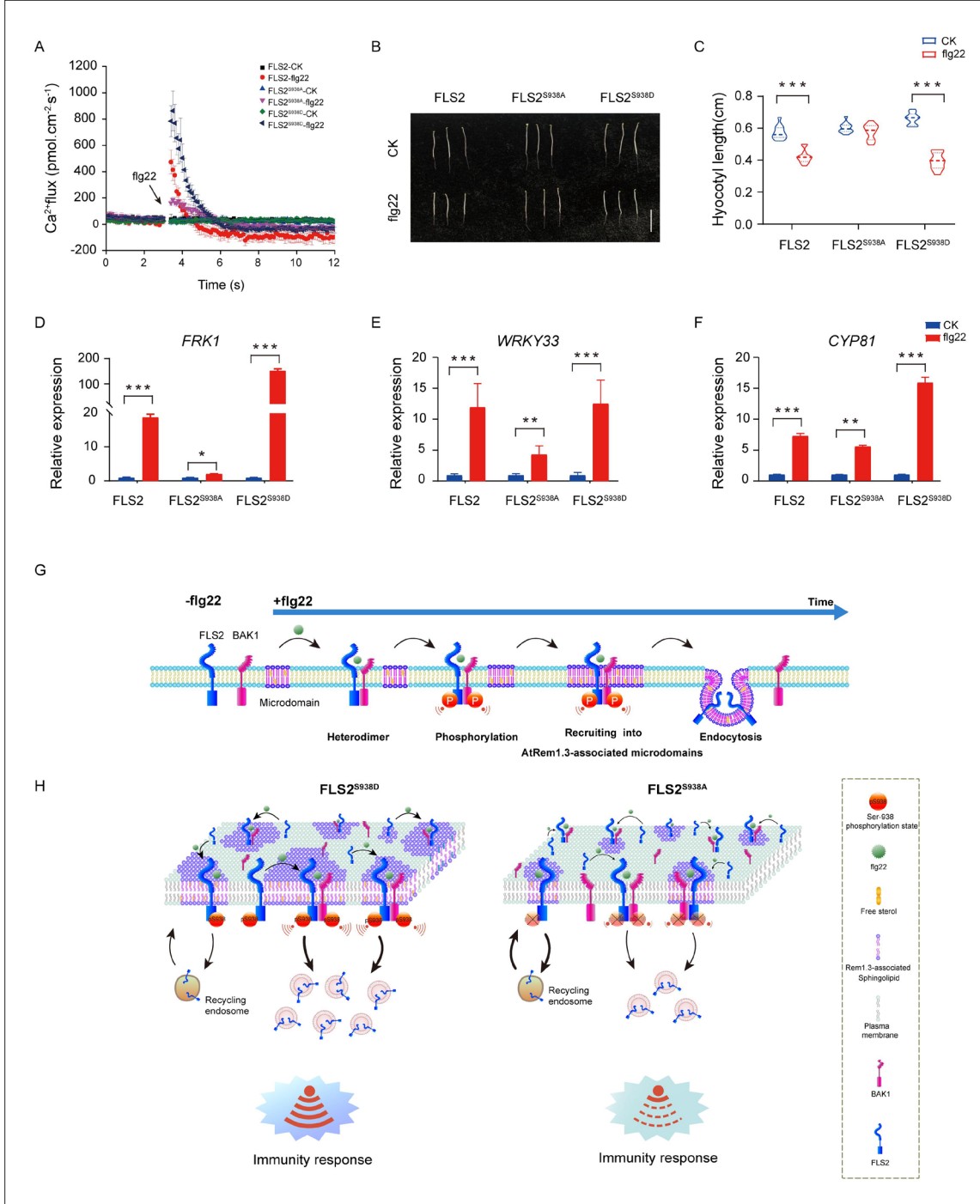

**Figure 4.** Ser-938 phosphorylation is essential for various flg22-induced pattern-triggered immunity (PTI) responses. (**A**) The flg22-induced transient $Ca^{2+}$ flux in 20-day-old transgenic leaf cells. The $Ca^{2+}$ flux was continuously recorded for 12 min in the test medium. Each point represents the average value for about 12 individual plants ± SEM. (**B**) Phenotypes of 5-day-old etiolated seedlings grown in the presence of 1/2 MS (control) or 10 µM flg22 solid medium. Bar = 0.5 cm. (**C**) Hypocotyl length of FLS2 (control, n = 13 seedlings; flg22, n = 13 seedlings), FLS2$^{S938A}$ (control, n = 13 seedlings; flg22, n = 13 seedlings), and FLS2$^{S938D}$ (control, n = 13 seedlings; flg22, n = 13 seedlings) transgenic plants. Statistical significance was assessed using Student's *t*-test (***p<0.001). Error bars represent the SD. (**D–F**) mRNA levels of the PTI marker genes *FRK1/WRKY33/CYP81* were significantly different between the FLS2, FLS2$^{S938A}$, and FLS2$^{S938D}$ 10-day-old transgenic *Arabidopsis* plant after treatment with 10 µM flg22 for 30 min. Total RNA was extracted from 10-day-old seedlings and analyzed using qRT-PCR, with transcript levels being normalized to *UBQ5*. Three biological replicates were performed. Each experiment was repeated thrice independently (n = 3). Statistical significance was assessed using Student's *t*-test (*p<0.05, **p<0.01, ***p<0.001). Error bars represent SD. (**G**) The working model for the spatiotemporal dynamic regulation of FLS2 phosphorylation at the plasma membrane upon flg22 stimulation. (**H**) Dynamic model of FLS2 with different Ser-938 phosphorylation states upon stimulation with flg22.

*Figure 4 continued on next page*

*Figure 4 continued*

The online version of this article includes the following source data and figure supplement(s) for figure 4:

**Source data 1.** The list of the transient Ca$^{2+}$ flux of FLS2/FLS2$^{S938A}$/FLS2$^{S938D}$-GFP under different treatments is shown in *Figure 4A*.

**Source data 2.** The list of the hypocotyl length of FLS2/FLS2$^{S938A}$/FLS2$^{S938D}$-GFP under different treatments is shown in *Figure 4C*.

**Source data 3.** The list of the mRNA levels of the PTI marker genes *FRK1/WRKY33/CYP81* of the FLS2, FLS2$^{S938A}$, and FLS2$^{S938D}$ 10-day-old transgenic *Arabidopsis* plants under different treatments is shown in *Figure 4D–F*.

**Figure supplement 1.** MAPKs phosphorylation in FLS2-GFP, FLS2$^{S938A}$-GFP, and FLS2$^{S938D}$-GFP 10-day-old seedlings incubated with 10 µM flg22 for 0 min, 5 min, and 15 min.

**Figure supplement 1—source data 1.** The original file for the western blot analysis.

**Figure supplement 1—source data 2.** The original file for the western blot analysis with highlighted bands and sample labels.

**Figure supplement 2.** Ser-938 phosphorylation affects flg22-induced callose deposition.

**Figure supplement 2—source data 1.** The list of the callose deposition of FLS2/FLS2$^{S938A}$/FLS2$^{S938D}$-GFP under different treatments.

**Figure supplement 3.** Phenotypes of 5-day-old etiolated seedlings grown in the 1/2 MS (CK) or solid medium with or without 10 µM flg22.

**Figure supplement 3—source data 1.** The original images of FLS2/FLS2$^{S938A}$/FLS2$^{S938D}$-GFP under different treatments.

encompass calcium burst activation, MAPKs cascade reaction, callose deposition, hypocotyl growth inhibition, and activation of immune-responsive genes. In contrast, FLS2$^{S938A}$ exhibited limited immune responses, underscoring the importance of Ser-938 phosphorylation for FLS2-mediated PTI responses.

In summary, our study confirmed that FLS2 phosphorylation regulated PAMP-triggered plant immunity by influencing spatiotemporal dynamics at the PM. Following flg22 treatment, activated FLS2 undergoes hetero-oligomerization and phosphorylation sequentially (*Figure 4G and H*). Crucially, phosphorylation at the Ser-938 site promotes FLS2 recruitment into *At*Rem1.3-associated nanodomains and endocytosis (*Figure 4G and H*). These results provided new insights into the phosphorylation-regulated dynamics of plant immunity, thereby providing a reference for future studies on signal transduction in intracellular complex nanostructures.

## Materials and methods

### Plant materials and construction

Mutants and transgenic lines used in all experiments were in the *Arabidopsis thaliana* Colombia-0 (Col-0) background. To generate the transgenic plants, specific constructs were PCR-amplified and cloned into the vector pCAMBIA2300. Plasmids were introduced into the mutant plants by *Agrobacterium*-mediated transformation. Dual-color lines expressing FLS2$^{S938A}$-GFP and FLS2$^{S938D}$-GFP with *At*REM1.3-mCherry were generated by hybridization.

### Transient infiltration of *N. benthamiana* leaves

The *Agrobacterium tumefaciens* strains GV3101 (obtained from Shanghai Weidi Biotechnology), harboring eukaryotic expression vectors, were infiltrated into the leaves of 8-week-old *N. benthamiana* plants. Plants were incubated at 22°C for 2 days before imaging.

### Drug treatments

All chemicals were obtained from Sigma-Aldrich and dissolved in 100% DMSO to yield stock solutions at the following concentrations: BFA (50 mM in DMSO, 50 µM in working solution), CHX (50 mM in DMSO, 50 µM in working solution), and FM4-64 (5 mM in DMSO, 5 µM in working solution). The flg22 of flagellin peptides was synthesized by Shanghai GL Biochem Company and was used at a concentration of 10 µM in double-distilled H$_2$O. *Arabidopsis* seedlings were treated in 1/2 MS growth liquid medium with added hormone or drug.

### Quantitative reverse-transcription PCR

Total RNA was extracted using the Plant Kit (Tiangen) and then reverse transcribed into cDNA with FastQuant RT Kit (Tiangen). Next, qPCR was performed using TiangenSurperRealPreMix Plus (SYBR Green). The primers used were as follows: WRKY33 (At At2g38470) 5′-GAAACAAATGGTGGGAATGG-3′ and 5′-TGTCGTGTGATGCTCTCTCC-3′; CYP81 (At At2g23190) 5′-AAATGGAGAGAGCAACACAA

TG-3′ and 5′-ATCGCCCATTCCAATGTTAC-3′; FRK1 (AtAt2g19190) 5′-TATATGGACACCGCGTATAG
TG-3′ and 5′-ATAAAACTTTGCGTTAGGGTCG-3′.

## Aniline blue staining

To detect callose deposition, aniline blue staining was performed as described. *Arabidopsis thaliana* leaves were completely de-colored by the destaining solution (3 ml ethyl alcohol and 1 ml glacial acetic acid), rinsed in water and 50% ethanol, and then stained in 150 mM $KH_2PO_4$ (pH 9.5) plus 0.01% aniline blue for 2 hr. Samples were mounted in 25% glycerol, and then observed under a microscope that was equipped with a Leica DM2500 UV lamp.

## Confocal laser scanning microscopy and image analysis

Confocal microscopy was done with a TCS SP5 Confocal Microscope fitted with a ×63 water-immersion objective. GFP and FM4-64 were assayed using 488 nm and 514 nm wavelengths (multitrack mode). The fluorescence emissions were respectively detected with spectral detector set LP 560-640 (FM4-64) and BP 520-555 (GFP). Image analysis was performed with Leica TCS SP5 software and quantified using the ImageJ software bundle (NIH).

## VA-TIRFM and single-particle fluorescence image analysis

The dynamics of FLS2 phospho-dead and phospho-mimic mutants were recorded using VA-TIRFM. This was done using an inverted microscope (IX-71, Olympus) equipped with a total internal reflective fluorescence illuminator (model no. IX2-RFAEVA-2; Olympus) and a 1003 oil-immersion objective (numerical aperture = 1.45). To track GFP-labeled proteins at the PM, living leaf epidermal cells of 6-day-old seedlings were observed under VA-TIRFM. To visualize GFP or mCherry fluorescent proteins, appropriate corresponding laser excitation (473 nm or 561 nm) was used and emission fluorescence was obtained with filters (BA510IF for GFP; HQ525/50 for mCherry). A digital EMCCD camera (Andor Technology, ANDOR iXon DV8897D-CS0-VP, Belfast, UK) was used to acquire the fluorescent signals, which were stored directly on computers and then analyzed using ImageJ software. Images of single particles were acquired with 100 ms exposure time.

## UMAP analysis

We used an R package to perform dimensionality reduction and clustering analysis of single-particle tracking data. We performed nonlinear dimensionality reduction for visualization with the function 'UMAP' in the R package. We proceeded to cluster cells using the Louvain algorithm with the 'Find-Neighbors' and 'FindClusters' functions in the R package carried out as previously described (*Dorrity et al., 2020*).

## TIRF-SIM imaging and the co-localization analysis

The SIM images were taken using a 60×NA 1.49 objective on a structured illumination microscopy (SIM) platform (DeltaVision OMX SR) with a sCMOS camera (camera pixel size, 6.5 µm). The light source for TIRF-SIM included diode laser at 488 nm and 568 nm with pixel sizes (µm) of 0.0794 and 0.0794 (*Barbieri et al., 2021*).

For the dual-color imaging, FLS2/FLS2^S938A/FLS2^S938D-GFP (488 nm/30.0%) and AtRem1.3-mCherry (561 nm/30.0%) were excited sequentially. The exposure time of the camera was set at 50 ms throughout single-particle imaging. The time interval for time-lapse imaging was 100 ms, the total time was 2 s, and the total time points were 21 s. The Imaris intensity correlation analysis plugin was used to calculate the co-localization ratio as described previously. The sizes of the ROIs used for statistical analysis are 13.38 µm and 13.38 µm.

## $Ca^{2+}$ flux measurements in *Arabidopsis* leaves

Net $Ca^{2+}$ fluxes in *Arabidopsis* leaf cells were measured using the non-invasive micro-test technique, as described previously (*Zhong et al., 2023*). First, a small incision was made in the leaves of 14-day-old seedling. Then it was fixed at the bottom of 35 mm Petri dish and incubated in the test buffer (pH = 6.0, 0.1 mmol $l^{-1}$ $KCl/CaCl_2/MgCl_2$, 0.2 mmol $l^{-1}$ $Na_2SO_4$, 0.3 mmol $l^{-1}$ MES, 0.5 mmol $l^{-1}$ NaCl) for approximately 30 min. The $Ca^{2+}$ concentration in the leaf cells was measured at 0.2 Hz near and 30 µm away from the cells. Each plant was measured once, and then using 1/2 MS (CK) or 10 µM flg22

treated the leaf and measured $Ca^{2+}$ flux again. The $Ca^{2+}$ flux was calculated as described (*Jiao et al., 2022*).

## Analysis of root growth

The transgenic seedlings were treated with different conditions, and imaging was performed by scanning the root systems at 500 dpi (Canon EOS 600D). ImageJ was used to analyze the root growth parameters. Three biological replicates were performed.

## Fluorescence recovery after photobleaching analysis

Plants were grown on 1/2 MS solid medium for 4 days prior to conducting FRAP analysis. The FRAP analysis was carried out using an Olympus FV1200 confocal microscope with an inverted microscope setup. The imaging was performed using 488 nm diode laser excitation and a water-immersed ×63 objective. The area of interest was bleached with a 488 nm laser at 100% laser power. The time interval for monitoring fluorescence recovery was 3 s. The fluorescence recovery data was subsequently analyzed using ImageJ and Origin 8.6 software following the methods outlined by *Xing et al., 2022*.

## Fluorescence correlation spectroscopy

FCS was performed in point-scanning mode on a Leica TCS SP5 FCS microscope equipped with a 488 nm argon laser, an Avalanche photodiode, and an in-house coupled correlator. After acquiring images on the PM of a cell in transmitted light mode, the diffusion of protein molecules into and out of the focal volume transformed the local concentration of fluorophores, leading to spontaneous fluctuation in the fluorescence intensity. Finally, the protein density was calculated on the basis of the protocol described previously.

## FRET-FLIM

FRET-FLIM analysis was performed using an inverted Olympus FV1200 microscope equipped with a Picoquant picoHarp300 controller. The excitation at 488 nm was implemented by a picosecond pulsed diode laser at a reduplication rate of 40 MHz by way of a water immersion objective (603, NA 1.2). The emitted light passed through a 520/35 nm bandpass filter and was detected by an MPD SPAD detector. Data were collected and performed using the SymphoTime 64 software (PicoQuant).

## Western blot

Total proteins were extracted from 10-day-old seedlings of the acidic phosphomimic mutants FLS2$^{S938A}$-GFP and FLS2$^{S938D}$-GFP and transgenic FLS2-GFP lines under different conditions. Proteins were extracted using buffer E (includes 1.5125 g Tris-HCl [pH 8.8], 1.2409 g $Na_2S_2O_5$, 11.1 ml glycerine, 1 g SDS, and 5 mM DTT). The proteins in PM fractions were obtained using the Invent Minute kit. Proteins were separated by 10% SDS-polyacrylamide gel and transferred to a nitrocellulose membrane. The membrane was blotted with anti-GFP antibody (Sigma-Aldrich) at a 1:4000 dilution.

## SR-Tesseler

The association of FLS2/FLS2$^{S938D}$/FLS2$^{S938A}$ with AtRem1.3-associated nanodomains was characterized using the open-source software packages ImageJ and SR-Tesseler. The specific operational steps were as previously described (*Levet et al., 2015*).

## PPI analysis

The degree of correlation between proteins was analyzed by measuring the PPIs (*Zinchuk et al., 2011*). Recently, an smPPI was developed. Using single-molecule images acquired via VA-TIRFM, the smPPI can be calculated to quantitatively assess the co-localization of FLS2/FLS2$^{S938D}$/FLS2$^{S938A}$ with AtRem1.3.

## Kymograph analysis

The original images were uploaded to ImageJ, which includes the 'multiple kymograph' plugin. The image can be adjusted for optimum contrast by selecting 'Image/Adjustment/Brightness/Contrast'. Subsequently, the individual particles of interest are selected using the 'straight line' tool, and the

dynamic features are analyzed using the 'Multiple Kymograph' tool. Finally, the 'Image/Type/RGB color' tool is utilized to obtain typical images, which are then saved in TIFF format.

## Pearson analysis

The co-localization analysis for FLS2/FLS2$^{S938D}$/FLS2$^{S938A}$ with AtRem1.3/BAK1 was carried out using ImageJ as described previously (*Zhang et al., 2019*). The background subtraction was performed using the 'Rolling Ball' method with a radius of 50 pixels. The plugin 'PSC co-localization' in ImageJ was employed to derive the Pearson correlation coefficient (Rr).

## Data analysis

The significance of arithmetic mean values for all data sets was assessed using Student's *t*-test. Error bars were calculated with the s.d. function in Microsoft Excel. The differences at p<0.05 were considered statistically significant. According to Student's *t*-test, characters in the figure represent statistically significant differences compared with control (*p<0.05, **p<0.01, and ***p<0.001).

## Acknowledgements

This work was supported by the National Natural Science Foundation of China (32030010, 91954202, 32370740, 32000483), National Key Research and Development Program of China (2022YFF071250, 2022YFF0712500, 2022YFD2200603), and Beijing Nova Program (20230484251).

## Additional information

### Funding

| Funder | Grant reference number | Author |
| --- | --- | --- |
| National Natural Science Foundation of China | 32030010 | Jinxing Lin |
| National Natural Science Foundation of China | 91954202 | Xiaojuan Li |
| National Natural Science Foundation of China | 32370740 | Yaning Cui |
| National Natural Science Foundation of China | 32000483 | Yaning Cui |
| National Key Research and Development Program of China | 2022YFF071250 | Jinxing Lin |
| National Key Research and Development Program of China | 2022YFF0712500 | Jinxing Lin |
| National Key Research and Development Program of China | 2022YFD2200603 | Jinxing Lin |
| Beijing Nova Program | 20230484251 | Yaning Cui |

The funders had no role in study design, data collection and interpretation, or the decision to submit the work for publication.

### Author contributions

Yaning Cui, Conceptualization, Writing - review and editing; Hongping Qian, Investigation, Writing - original draft; Jinhuan Yin, Data curation, Visualization, Methodology; Changwen Xu, Formal analysis, Visualization; Pengyun Luo, Methodology; Xi Zhang, Writing - review and editing; Meng Yu, Data curation; Bodan Su, Validation; Xiaojuan Li, Software; Jinxing Lin, Resources, Funding acquisition

### Author ORCIDs

Hongping Qian (iD) http://orcid.org/0009-0003-7332-6503

Jinxing Lin https://orcid.org/0000-0001-9338-1356

Reviewer #1 (Public Review): https://doi.org/10.7554/eLife.91072.3.sa1
Reviewer #2 (Public Review): https://doi.org/10.7554/eLife.91072.3.sa2
Author response https://doi.org/10.7554/eLife.91072.3.sa3

## Additional files

### Supplementary files
• MDAR checklist

### Data availability
All data are included in the manuscript and supporting files.

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
