## [Editor Report · eLife assessment]

This potentially **important** study employs advanced imaging techniques to directly visualize molecular dynamics and of the immune receptor kinase FLS2 in specific microenvironments. The evidence supporting the ligand-induced association with remorin and the requirement of a previously reported phosphosite as presented is **solid**, although support by independent methods would be welcome. The work will be of interest to plant biologists working on cell surface receptors.

---

## [Referee Report · Reviewer #1 (Public Review)]

Summary:

Organization of cell surface receptors in membrane nanodomains is important for signaling, but how this is regulated is poorly understood. In this study the authors employ TIRFM single-molecule tracking combined with multiple analyses to show that ligand exposure increases diffusion of the immune receptor FLS2 in the plasma membrane and its co-localization with remorin REM1.3 in a manner dependent on the phosphosite S938. They additionally show that ligand increases dwell time of FLS2, and this is linked to FLS2 endocytosis, also in a manner dependent on S938 phosphorylation. The study uncovers a regulatory mechanism of FLS2 localization in the nanodomain crucial for signaling.

Strengths:

TIRFM single-molecule tracking, FRAP, FRET and endocytosis experiments were nicely done. A role of S938 phosphorylation is convincing.

Weaknesses:

In the previous submission, reviewers pointed out multiple issues, which the reviewers believed the authors can address in the revision. The revised version does improve to some extent but still contains many issues in terms of data analysis and writing.

---

## [Referee Report · Reviewer #2 (Public Review)]

Summary:

The research conducted by Yaning Cui and colleagues delves into understanding FLS2-mediated immunity. This is achieved by comparing the spatiotemporal dynamics of a FLS2-S938A mutant and FLS2-WT, especially in relation to their association with the remorin protein. To delineate the differences between the FLS2-S938A mutant and FLS2-WT, they utilized a plethora of advanced fluorescent imaging techniques. By analyzing surface dynamics and interactions involving the receptor signal co-receptor BAK1 and remorin proteins, the authors propose a model of how FLS2 and BAK1 are assembled and positioned within a remorin-specific nano-enviroment during FLS2 ligand-induced immune responses.

Strengths:

These techniques offer direct visualizations of molecular dynamics and interactions, helping us understand their spatial relationships and interactions during innate immune responses.

Advanced cell biology imaging techniques are crucial for obtaining high-resolution insights into the intracellular dynamics of biomolecules. The demonstrated imaging systems are excellent examples to be used in studying plant immunity by integrating other functional assays.

Weaknesses:

It's essential to acknowledge that every fluorescence-based method, just like biochemical assays, comes with its unique limitations. These often pertain to spatial and temporal resolutions, as well as the sensitivity of the cameras employed in each setup. Meticulous interpretation is pivotal to guarantee an accurate depiction and to steer clear of potential misunderstandings when employing specific imaging systems to analyze molecular attributes. Moreover, a discerning interpretation and accurate image analysis can offer invaluable guidance for future studies on plant signaling molecules using these nice cell imaging techniques.

For instance, although single-particle analysis couldn't conclusively link FLS2 and remorin, FLIM-FRET effectively highlighted their ligand-triggered association and the disengagement brought on by mutations. While these methodologies seemed to present differing outcomes, they were described in the manuscript as harmonious. In reality, these differences could highlight distinct protein populations active in immune responses, each accentuated differently by the respective imaging techniques due to their individual spatial and temporal limitations. Addressing these variations is imperative, especially when designing future imaging explorations of immune complexes.

---

## [Author Response]

The following is the authors’ response to the original reviews.

We greatly thank you and the reviewers for your expert comments and valuable suggestions on our manuscript. After reading these comments, we realized that the previous version of the manuscript contained some weak points. Surely, the issues raised by the six reviewers were of great help in the revision of our manuscript.

According to the comments, we have now fully revised the manuscript to address most of the questions and suggestions. In addition, we reworded some parts of the Introduction, Results and Discussion, Figures, Figure legends and Experimental Methods to increase the rigor of our conclusions.

Overall, you will see that we have paid serious attention to all the concerns and criticisms expressed by reviewers. Addressing these various issues has most certainly allowed us to prepare a much-improved manuscript and for this we offer our hearty thanks.

**Reviewer #1 (Public Review):**
Summary:The organization of cell surface receptors in membrane nanodomains is important for signaling, but how this is regulated is poorly understood. In this study, the authors employ TIRFM single-molecule tracking combined with multiple analyses to show that ligand exposure increases the diffusion of the immune receptor FLS2 in the plasma membrane and its co-localization with remorin REM1.3 in a manner dependent on the phosphosite S938. They additionally show that ligand increases the dwell time of FLS2, and this is linked to FLS2 endocytosis, also in a manner dependent on S938 phosphorylation. The study uncovers a regulatory mechanism of FLS2 localization in the nanodomain crucial for signaling.Strengths:TIRFM single-molecule tracking, FRAP, FRET, and endocytosis experiments were nicely done. The role of S938 phosphorylation is convincing.Weaknesses:Question 1: The model suggests that S938 is phosphorylated upon flg22 treatment. This is actually not known.

Reply: Thank you for your expert comments. Although the phosphorylation of Ser-938 upon flg22 treatment is not known, the model presented in the manuscript is based on previous studies that have shown the importance of Ser-938 phosphorylation for the function of FLS2 (Cao et al, 2013). When it is mutated to the phosphorylation-mimicking residues aspartate or glutamate, immune responses remain normal. These findings suggest that the phosphorylation of Ser-938 plays a critical role in activating defense mechanisms upon flagellin detection (Cao et al, 2013). Now we added the results of Cao et al. (2013) to the introduction to strengthen in the revised manuscript.

Question 2: In addition, the S938D mutant does not show constitutively increased diffusion and co-localization with remorin. It is necessary to soften the tone in the conclusion.Reply: We appreciate the valuable suggestions from the reviewer. Based on our findings, we observed that the phosphorylation of Ser-938 significantly impacts the dynamics of flg22-induced FLS2. However, it does not alter the diffusion coefficient of FLS2 itself. In the revised manuscript, we have carefully adjusted the conclusion by softening the tone to reflect these findings.Question 3: The introduction (only two paragraphs) and discussion are not properly written in the context of the current understanding of plant receptors in nanodomains. The authors basically just cited a few publications of their own, and this is not acceptable.

Reply: We accepted the criticisms here. Now, we have reworded the introduction and discussion sections to improve clarity. Furthermore, we have incorporated several new reports on plant receptors in nanodomains into the revised manuscript. Besides, we deleted some publications from our own group, while citing the latest references on plant receptors and nanodomains.

**Reviewer #2 (Public Review):**
Summary:The research conducted by Yaning Cui and colleagues delves into understanding FLS2-mediated immunity. This is achieved by comparing the spatiotemporal dynamics of an FLS2-S938A mutant and FLS2-WT, especially in relation to their association with the remorin protein. To delineate the differences between the FLS2-S938A mutant and FLS2-WT, they utilized a plethora of advanced fluorescent imaging techniques. By analyzing surface dynamics and interactions involving the receptor signal co-receptor BAK1 and remorin proteins, the authors propose a model of how FLS2 and BAK1 are assembled and positioned within a remorin-specific nano-environment during FLS2 ligand-induced immune responses.Strengths:These techniques offer direct visualizations of molecular dynamics and interactions, helping us understand their spatial relationships and interactions during innate immune responses. Advanced cell biology imaging techniques are crucial for obtaining high-resolution insights into the intracellular dynamics of biomolecules. The demonstrated imaging systems are excellent examples to be used in studying plant immunity by integrating other functional assays.Weaknesses:It's essential to acknowledge that every fluorescence-based method, just like biochemical assays, comes with its unique limitations. These often pertain to spatial and temporal resolutions, as well as the sensitivity of the cameras employed in each setup. Meticulous interpretation is pivotal to guarantee an accurate depiction and to steer clear of potential misunderstandings when employing specific imaging systems to analyze molecular attributes. Moreover, a discerning interpretation and accurate image analysis can offer invaluable guidance for future studies on plant signaling molecules using these nice cell imaging techniques. For instance, although single-particle analysis couldn't conclusively link FLS2 and remorin, FLIM-FRET effectively highlighted their ligand-triggered association and the disengagement brought on by mutations. While these methodologies seemed to present differing outcomes, they were described in the manuscript as harmonious. In reality, these differences could highlight distinct protein populations active in immune responses, each accentuated differently by the respective imaging techniques due to their individual spatial and temporal limitations. Addressing these variations is imperative, especially when designing future imaging explorations of immune complexes.

Reply: Thank you for your insightful comments and suggestions. We appreciate your expertise in fluorescence-based methods and the importance of careful interpretation and accurate image analysis. We agree with you that different imaging techniques may have their limitations and can highlight distinct aspects of protein dynamics and interactions.

In our study, we used single-particle analysis and FLIM-FRET to investigate the spatiotemporal dynamics of FLS2 and its association with remorin. While single-particle analysis did not conclusively link FLS2 and remorin, FLIM-FRET effectively highlighted their ligand-triggered association and the disengagement caused by mutations. We acknowledge that these techniques may have different spatial and temporal resolutions, leading to the discrepancy in their results. However, after the normalized treatment, we can provide very similar conclusions. Accordingly, we have revised the manuscript.

**Reviewer #3 (Public Review):**
Summary:Receptor kinases (RKs) perceive extracellular signals to regulate many processes in plants. FLS2 is an RK that acts as a pattern-recognition receptor (PRR) to recognize bacterial flagellin and activate pattern-triggered immunity (PTI). PRRs such as FLS2 have been previously shown to reside within PM nanodomains, which can regulate downstream PTI signaling. In the current manuscript, Cui et al use single particle tracking to characterize the effect of previously-described phosposite mutants (FLS2-S938A/D) on the PM organization, endocytosis, and signaling functions of FLS2. The authors confirm that FLS2-S938D but not -S938A is functional for flg22-induced responses, while also demonstrating that phopshodead mutation at this site (S938A) prevents flg22-induced sorting into nanodomains and endocytosis. These results are consistent with S938 being an important phosphorylation site for FLS2 function, however, they fall short of demonstrating that membrane disorganization of FLS2-938A is responsible for downstream signaling defects.Strengths:The authors' experiments (single particle tracking, co-localization, etc) do a good job of demonstrating how a non-functional version of FLS2 (S938A) does not alter its spatio-temporal dynamics, nanodomain organization, and endocytosis in response to flg22, suggesting that these require a functional receptor and are regulated by intracellular signaling components.Weaknesses:Question 1: The authors do not provide direct evidence that S938 phosphorylation specifically affects membrane organization, rather than FLS2 signaling more generally. All evidence is consistent with S938A being a non-functional version of FLS2, wherein an activated/functional receptor is required for all downstream events including membrane re-organization, downstream signalling, internalization, etc. Furthermore, the authors never demonstrate that this site is phosphorylated in planta in the basal or flg22-elicited state.

Reply: Sorry that we did not describe clearly in the original manuscript. In fact, we found in our study that the phosphorylation of the Ser-938 site influences the efficient sorting of FLS2 into AtRem1.3-associated microdomains rather than membrane organization, as depicted in Figure 2. Furthermore, we found that the immune responses are disrupted when Ser-938 is mutated to alanine, which is consistent with previously reported results (Cao et al, 2013). However, they remain normal when mutated to the phosphorylation-mimicking residues aspartate or glutamate. These results suggest that the phosphorylation of Ser-938 is crucial for activating defense mechanisms upon flagellin detection. Although the phosphorylation of Ser-938 in plant at the basal or flg22-elicited state is not known, the model presented in the manuscript is based on the results of our current investigation together with those in the previous study that have shown the importance of Ser-938 phosphorylation for FLS2 function (Cao et al, 2013).

Question 2: As written, the manuscript also has numerous scientific issues, including a misleading/incomplete description of plant immune signaling, lack of context from previous work, and extensive use of inappropriate references.

Reply: We accept the criticism here. After reading the comments, we realized the problem. Now we have revised the misleading or incomplete description of plant immune signaling, added the context of previous works and deleted inappropriate references in the revised manuscript.

**Reviewer #1 (Recommendations For The Authors):**
Question 1: The description of the data has no details. How many biological repeats were done? How were statistical analyses done? What is the concentration of flg22? How was the calcium flux done (Fig. 4A)? The method also lacks details and relevant references.

Reply: We apologize for the lack of detail in presenting the data. Following your suggestion, we added comprehensive figure legends that provide clear explanations for each figure. Additionally, we included supplementary information on the measurement methods and references pertaining to calcium flux in the revised manuscript.

Question 2: Data in Fig. 4 basically repeated the 2013 PLoS Pathog paper. Why were these experiments even performed? Were GFP-tagged FLS2 lines used in these experiments? If this is the case, the data just verified that the GFP-tagged FLS2 functions as expected and should be moved to supporting data.

Reply: Thanks for the expert suggestions. In our study, we utilized GFP-tagged FLS2 lines to generate FLS2-S938 mutants and conducted experiments to investigate the flg22-induced immune response. Although some experiments in Figure 4 are similar to those reported (Cao et al, 2013), we provided a more detailed analysis of the immune response. The comprehensive analysis included early immune responses and late immune responses, e.g., the activation of a calcium burst, mitogen-activated protein kinases (MAPKs), the induction of immune-responsive genes and callose deposition, ultimately resulting in the inhibition of plant growth. As some results are analogous to the previous paper, we transfer some of the experiments as suggested, including the analysis of MAPKs and callose deposition, to the supporting data section of the revised manuscript.

Question 3: Flg22-induced FLS2-BAK1 association does not require S938, this is consistent with prior study that flg22 acts as a molecular glue for the ectodomains of FLS2 and BAK1 (Sun et al., 2013 Science). This needs to be cited.

Reply: Yes, we agree with the comment. Now we added an additional sentence in the revised manuscript: “ This aligns with the previous finding that flg22 acts as a molecular glue for FLS2 and BAK1 ectodomains (Sun et al., 2013).”

Question 4: Line 50, the references cited do not match what they say here.

Reply: We are sorry for the mistake in citing inappropriate references. In the revised manuscript, we deleted this sentence as well as the incorrect reference.

Question 5: Line 105, "flg22 can act as a ligand-like factor". It is a ligand!

Reply: Sorry for the mistake. Now, the sentence was corrected in the revised manuscript by deleting the word “like”.

Question 6: Line 107, FLS2/BAK1 heterodimerization, not heteroologomerization.

Reply: Now we used “heterodimerization” to replace “heteroologomerization” in the revised manuscript.

Question 7: Line 114, are these really the best references to cite here?

Reply: After reading the comment, we found the references were not suitable here. Now we changed references by citing “(Martinière et al., 2021)” in the revised manuscript.

Question 8: Lines 123-124, the sentence is incomplete.

Reply: In the revised manuscript, we reworded the sentence to make it complete now. We changed “In a previous investigation, we demonstrated that flg22 induces FLS2 translocation from AtFlot1-negative to AtFlot1-positive nanodomains in the plasma membrane, implying a connection between FLS2 phosphorylation and membrane nanodomain distribution (Cui et al., 2018). To validate this, we assessed the association of FLS2/FLS2S938D/FLS2S938A with membrane microdomains, using AtRem1.3-associated microdomains as representatives (Huang et al., 2019).” in the revised manuscript.

Question 9: Lines 169-170, Why is this "most important"?

Reply: Sorry for the unsuitable description. As we have dramatically changed the manuscript, this sentence was deleted from the new version.

**Reviewer #2 (Recommendations For The Authors):**
Here are some specific areas of ambiguity in the study to be improved.Question 1: Clarity in statistical analysis is necessary. Many figure legends omit details such as the sample size "n", and the nature of the measurements, like ROIs, images, and dots, the size of the seedlings, etc.

Reply: We appreciated this suggestion, which was raised by the reviewer I as well. Now, we provided the details for each figure, including the sample size, the nature of the measurements in the revised manuscript.

Question 2: Additional background about the choice of FLS2-S938 mutant would be beneficial, given that this mutant doesn't affect the BAK1 interaction but nullifies several PTI responses.

Reply: Yes, we agreed that some additional background is required for the FLS2-S938 mutant. Therefore, we added a sentence here: “FLS2 Ser-938 mutations impact flg22-induced signaling, while BAK1 binding remains unaffected, thereby suggesting Ser-938 regulates other aspects of FLS2 activity (Cao et al., 2013).” in the revised manuscript.

Question 3: A specific segment "... Using CLSM, Fluorescence Correlation Spectroscopy (FCS) and Western blotting, we found that the endocytic vesicles of FLS2S938D increased significantly after flg22 treatment (Figure 3B-3E)..." is not easy to follow. The author may want to differentiate these methods and highlight them by indicting them as endocytic vesicle counting, receptor density on PM measurement by FCS, and WB-based protein degradation characterization to understand such mixed descriptions better. By the way, "Number of Endocytosis" should be "number of endocytic vesicles". Endocytosis is a process and uncountable.

Reply: We thank the reviewer for kindly reminding us to differentiate experimental methods. Therefore, we changed the sentences in the revised manuscript: “Employing confocal laser-scanning microscopy (CLSM) during 10μM flg22 treatment, we tracked FLS2 endocytosis and quantified vesicle numbers over time (Figure 3B). It is evident that both FLS2 and FLS2S938D vesicles appeared 15 min after-flg22 treatment, significantly increasing thereafter (Figure 3C). Notably, only a few vesicles were detected in FLS2S938A-GFP, indicating Ser-938 phosphorylation's impact on flg22-induced FLS2 endocytosis. Additionally, fluorescence correlation spectroscopy (FCS) (Chen et al., 2009) monitored molecular density changes at the PM before and after flg22 treatment (Figure S3F). Figure 3D shows that both FLS2-GFP and FLS2S938D-GFP densities significantly decreased after flg22 treatment, while FLS2S938A-GFP exhibited minimal changes, indicating Ser-938 phosphorylation affects FLS2 internalization. Western blotting confirmed that Ser-938 phosphorylation influences FLS2 degradation after flg22 treatment (Figure 3E), consistent with single-molecule analysis findings.” Besides, we also changed “number of endocytosis” to “the number of endocytic vesicles” in Figure 3C as suggested.

Question 4: In Figure 1 E, a discrepancy exists where the total percentages in the red and black columns don't sum up to 100%, while other groups look right. This needs clarification.

Reply: We are sorry for our carelessness in making the data incomplete. Now we thoroughly supplemented, collated, and rechecked the data in Figure 1E. Due to an oversight during the production of the figure, some data was inadvertently omitted, resulting in the red column not reaching 100%. Besides, we checked the data in the black column again, and the total percentage indeed added up to 100%.

Question 5: Although Figure 1F uses UMAP analysis to differentiate between FLS2WT and A mutants, only data pertaining to the "D" mutant is shown.

Reply: Thank you for the expert comments. Because there are several images in Figure 1, we only selected the data related to the “D” mutant as a representative for display. As suggested, we have added all the UMAP images in the revised supplement figure S1F.

Question 6: There are apparent inconsistencies in the FRAP results, particularly regarding the initial recovery points post-bleaching. A detailed statistical analysis, supplemented with FRAP images over time, should be included for clarity. Were they bleached to a similar ground level before monitoring their recovery? The data points from "before" and "after "bleaching were not shown. I found the red and blue curves showed similar recovery slop, which suggests no long-distance movement changes for all three FLS2 versions, with or without flg22. This is opposite from the conclusions made by the author.

Reply: Thank you for the expert comments. After reading the comments, we recognized this terrible problem. Therefore, we carried out a new FRAP experiment. The new results showed that, following complete bleaching of three samples of FLS2 to ground level, the recovery rates of FLS2 and FLS2S938D under flg22 treatment were significantly higher compared to the control group (Fig. 1G). In contrast, the recovery rates of the FLS2S938A-GFP after flg22 treatment remain similar to that before treatment (Fig. 1G), indicating that the Ser-938 phosphorylation site indeed affects the flg22-induced lateral diffusion of FLS2 at the PM. The new results are basically consistent with the motion range of single-molecule results, which is not contradictory to long-distance movement changes. Accordingly, we incorporated the new time-lapse FRAP images into Figure 1G and S1B.

Question 7: There's a potential typo in Figure 1B regarding the bar size. It could neither possibly be 200 um nor 200 nm. Figure 1A also needs a scale bar.

Reply: Apologies for the mistake. We now corrected “200 μm” to “2 μm”. Besides, we also included a scale bar in Figure 1A in the revised manuscript.

Question 8: Due to the unreliable tracking for a long-time by Imaris, the authors analyzed the tracks within 10s and quantified very short live particles under 4s. Such 4S surface retention for a receptor does not seem to match functional endocytic internalization time for cargo. Even after the endocytic adaptor module recruitment, it would take at least more than 10s to finish the internalization. In the field of endocytosis, these events are often described as abortive endocytic events. However, the disappearance of cargoes, FLS2 in this case, indicates internalization into the cytoplasm, which is interesting. May the author discuss more on how these short events analyzed enhance our understanding of the functional behavior of FLS2?

Reply: We greatly appreciated the valuable comments provided by the reviewer. After thorough consideration, we acknowledged that in our original manuscript, we failed to distinguish the short-lived from the long-lived particles and vaguely put them collectively into the internalized particles. We realized that and it is inappropriate to ambiguously categorize all particles as internalized. Therefore, we added the sentence “Additionally, numerous FLS2 exhibited short-lived dwell times, indicating abortive endocytic events associated with the endocytic pathway and signal transduction (Bertot et al., 2018)” in the revised manuscript.

Question 9: Figure 2D should be comprehensive, presenting data for the WT, A, and D versions.

Reply: Yes, we agreed with the suggestions. Now, we added several representative images for the WT, A, and D versions in the revised manuscript.

Question 10: In Figure 2D, TIRM-SIM should be a typo and rectified to TIRF-SIM. Also, a detailed explanation of the TIRF-SIM setup and its specifics would be important. The imaging approach of SIM, especially the time duration for finishing all frames before reconstruction, is essential to rationalize its use in capturing and measuring an appropriate speed range of particle movement. May the author elaborate on the technique details and the use of TIRF-SIM for colocalization analysis? To clarify these, the author may provide additional TIRF-only movies of FLS2 (WT, A, D) and AtRem1.3 for comparison with TIRF-SIM still images.

Reply: Sorry for the mistake. In the revised manuscript, we have corrected “TIRM-SIM” to “TIRF-SIM”. In order to rationalize its use in capturing and measuring an appropriate speed range of particle movement, we included a more detailed description of the imaging approach and the colocalization analysis of TIRF-SIM in the Materials and Methods section as follows: “The SIM images were taken by a 60 × NA 1.49 objective on a structured illumination microscopy (SIM) platform (DeltaVision OMX SR) with a sCMOS camera (Camera pixel size, 6.5 μm). The light source for TIRF-SIM included diode laser at 488 nm and 568 nm with pixel sizes (μm) of 0.0794 and 0.0794 (Barbieri et al., 2021). For the dual-color imaging, FLS2/FLS2S938A/FLS2S938D-GFP (488 nm/30.0%) and AtRem1.3-mCherry (561 nm/30.0%) were excited sequentially. The exposure time of the camera was set at 50 ms throughout single-particle imaging. The time interval for time-lapse imaging was 100 ms, the total time was 2s, and the total time points were 21s. The Imaris intensity correlation analysis plugin was used to calculate the co-localization ratio.” in the revised manuscript. Furthermore, we provided additional TIRF-SIM movies of FLS2 (WT, A, D) and AtRem1.3.

Question 11: The colocalization displayed in Figure 2D is hard to tell. A colocalization ratio of FLS2-AtRem1.3 is shown as ~0.8%, which has only ~0.2% difference from the flg22-treated condition. "n" of Figure 2F should be specified in the legend, such as a line with a specific length, or an ROI with a specific area size.

Reply: Thank you for the expert comments. Although the increased colocalization after flg22 treatment is not high, the change is statistically significant as compared with the wild type. We agreed that every fluorescence-based method, like biochemical analysis, has its own unique limitations, which were raised by the Reviewer #2 (Public Review) as well. In order to provide strong evidence, we also carried out the FLIM-FRET experiment as a supplement, which can effectively detect their ligand-triggered association or disassociation. From figure 2G and H, we clearly found that the co-localization of FLS2/FLS2S938D-GFP with AtRem1.3-mCherry significantly increase in response to flg22 treatment (FLS2-GFP control: 2.45 ± 0.019 s; FLS2-GFP flg22-treated: 2.39 ± 0.016 s; FLS2S938D-GFP control: 2.42 ± 0.010 ns; FLS2S938D-GFP flg22-treated: 2.35 ± 0.028 ns). In contrast, FLS2S938A-GFP shows no significant changes (control: 2.53 ± 0.011 ns; flg22-treated: 2.56 ± 0.013 ns), indicating that Ser-938 phosphorylation influences efficient sorting of FLS2 into AtRem1.3-associated microdomains. Following the suggestion of the reviewer, we now rearranged the order of 2E and 2F, in which N represents the entire image region used for analysis rather than a specific region of interest.

Question 12: I appreciate the nice results of the FLIM-FRET results for FLS2-Rem1.3. Figure 2H should be supplemented with additional representative images of all FLS2 variants including WT and mutants.

Reply: Thanks for your warm encouragement. As suggested, we added all the representative images in the revised manuscript.

Question 13: The unit of the X-axis of Figure 2E can not be pixel. Should it be, um? In the method, the author could specify the camera model and magnification for TIRF-SIM to understand pixel size of the image better.

Reply: Sorry for the mistake here. Indeed, the unit of the X-axis in Figure 2E should be μm. Now we correct this mistake in Figure 2E in the revised manuscript. Besides, we included a detailed description of the imaging approach of TIRF-SIM in the Materials and Methods section as follows: “The SIM images were taken by a 60 × NA 1.49 objective on a structured illumination microscopy (SIM) platform (DeltaVision OMX SR) with a sCMOS camera (Camera pixel size, 6.5 μm)”.

Question 14: "... as shown in A..." in Figure Legend 2E should be "... as shown in D..."

Reply: Thanks for pointing out this mistake. In the revised manuscript, we used “as shown in D” to replace “as shown in A”.

Question 15: I recommend that the authors exercise caution when drawing conclusions based on the Rem1.3 data and when representing the "microdomain" concept in their final model. While Rem1.3 punctate is a nanometer-sized protein cluster specific to its identity, its shape can be categorized as a nanodomain. Conceptually, however, it neither universally represents all nanodomains nor microdomains, as depicted in Figure 4. We should exercise caution to prevent providing misleading information to the field.

Reply: We thank the reviewer for expert comments. To avoid misleading conclusions, we changed “nanodomains” to “AtRem1.3-associated microdomains” in the revised manuscript. Besides, we have also made modifications to Figure 4.

**Reviewer #3 (Recommendations For The Authors):**
Question 1: The manuscript needs to be extensively re-written and has severe issues as-is. Many references are either not quite appropriate or are completely unrelated to the use in the text. In general, the current state-of-the-art of PTI and RK signaling is not correctly described or incorporated.

Reply: We accepted the criticisms here. As suggested, we thoroughly rewrote the manuscript to address the concerns raised. Furthermore, we have thoroughly checked and revised the manuscript by removing 21 irrelevant references and adding 30 relevant references. We also incorporated the most up-to-date descriptions of the PTI and RK signaling pathways.

Question 2: Receptor-like kinase (RLK) should generally be receptor kinase (RK) as receptor functions are now well established.

Reply: Yes, we agreed with your expert comment here. Now, we changed “Receptor-like kinase (RLK)” into “receptor kinase (RK)” in the revised manuscript.

Question 3: Line 20 - is this really true?

Reply: Sorry for the mistake. In the revised manuscript, we changed “However, the mechanisms underlying the regulation of FLS2 phosphorylation activity at the plasma membrane in response to flg22 remain largely enigmatic.” to “However, the dynamic FLS2 phosphorylation regulation at the plasma membrane in response to flg22 needs further elucidation.”

Question 4: S938D sorts better in response to Flg22; S938A is unaffected - suggests phosphorylation of S938 is not dynamic in response to Fig 22 but is required for pre-elicitation sorting. Overall, there is a chicken-and-egg problem in this paper: which comes first, immune/signalling functionality or nanodomain sorting? And which is explaining the defects of S938A?

Reply: We thank the reviewer for expert suggestions. In fact, the previous studies showed that membrane microdomains serve as signaling platforms that mediate cargo protein sorting and protein-protein interactions in a variety of contexts (Goldfinger et al. 2017). Since our previous research showed that the disruption of membrane microdomains affected flg22-induced immune signaling (Cui et al. 2018), we speculate that the immune signal occurred after entering the membrane microdomains.

As shown in Figure 1 and 2, ligand exposure leads to an increase in diffusion coefficient and enhanced co-localization with REM1.3, both of which are dependent on the phosphorylation of the Ser-938 site. Deducing from these results, we inferred that the defects in S938A resulted largely from its failure to sort into membrane microdomains. The phosphorylation of the Ser-938 site can regulate FLS2 into functional AtRem1.3-associated microdomains, thereby affecting flg22-induced plant immunity.

Question 5: Line 37 conserved, not conservative (though not technically true - the domain organization is conserved but the ECDs are not conserved).

Reply: Thank you for pointing this mistake out. In the revised manuscript, we used “conserved” to replace “conservative”.

Question 6: Lines 40-42 - not all phosphorylation sites are within the kinase domain, for example, sites are well-described on the JM and/or C-tail regions outside of the kinase domain.

Reply: We accepted the criticisms here. We have corrected the sentence to “with phosphorylation sites mainly located in PKC” in the revised manuscript.

Question 7: Line 42 - what is BIK1? Intro to relevant topics is severely lacking.

Reply: Sorry for the incomplete introduction here. We added the relevant introduction of BIK1 by adding that “Upon recognizing flg22, FLS2 interacts with the co-receptor Brassinosteroid-Insensitive 1-associated Kinase 1 (BAK1), initiating phosphorylation events through the activation of receptor-like cytoplasmic kinases (RLCKs) such as BOTRYTIS-INDUCED KINASE 1 (BIK1) to elicit downstream immune responses (Chinchilla et al., 2006; Li et al., 2016b; Majhi et al., 2021). ” in the revised manuscript.

Question 8: Lines 42-44 - not sure this sequence of events is being properly described (e.g. BIK1 release is unlikely to precede activation by BAK1/SERKs).

Reply: We apologize for not expressing this sentence clearly. Now, we reworded the sentence: “Upon recognizing flg22, FLS2 interacts with the co-receptor Brassinosteroid-Insensitive 1-associated Kinase 1 (BAK1), initiating phosphorylation events through the activation of receptor-like cytoplasmic kinases (RLCKs) such as BOTRYTIS-INDUCED KINASE 1 (BIK1) to elicit downstream immune responses (Chinchilla et al., 2006; Li et al., 2016b; Majhi et al., 2021).” in the revised manuscript.

Question 9: Line 61 - S938 was identified by Cao et al (2013) based on in vitro MS, but was functionally validated using genetic assays, not based on MS.

Reply: Thank you for your comments. Now, we changed the sentence: “In vitro mass spectrometry (MS) identified multiple phosphorylation sites in FLS2. Genetic analysis further identified Ser-938 as a functionally important site for FLS2 in vivo (Cao et al., 2013).” in the revised manuscript.

Question 10: Line 68-69 - phospho-dead and phospho-mimic, not phosphorylated/non-phosphorylated.

Reply: We thank the reviewer for expert suggestions. In the revised manuscript, we changed the sentence by replacing “phosphorylated/non-phosphorylated” with “phospho-mimic” and “phospho-dead”.

Question 11: Lines 104-106 - this is wildly misleading. Flg22 is more than a ligand-like factor, as it is a bona fide ligand, and the heterodimerization with BAK1/SERKs is extremely well-established (and relevant foundational papers should be cited here in place of the authors' previous work).

Reply: We apologize for the incorrect expression here. After reading the comments, we realized the problem which was raised by the reviewer I as well. Now, we changed “ligand-like factor” to “ligand”. Besides, we cited the new references “(Orosa et al., 2018)” to replace the references of our group in the revised manuscript.

Question 12: Lines 107-112 - again, this is confusing. There is a decade of (uncited, undiscussed) work previously establishing that heterodimerization of RK-co-receptor complexes is mediated by extracellular ligand binding and independent of intracellular phosphorylation.

Reply: We thank the reviewer for expert suggestions. Now, we added several sentences in the revised manuscript: “Therefore, we further investigated if Ser-938 phosphorylation affects FLS2/BAK1 heterodimerization. Tesseler segmentation, FRET-FLIM, and smPPI analyses revealed no impact of Ser-938 phosphorylation on FLS2/BAK1 heterodimerization (Figure 2A-C and S2). This aligns with the previous finding that flg22 acts as a molecular glue for FLS2 and BAK1 ectodomains (Sun et al., 2013), confirming the independence of FLS2/BAK1 heterodimerization from phosphorylation, with these events occurring sequentially.”

Question 13: Line 119 - this is the wrong citation - Yu et al 2020 is a review and does not cover RALFs; correct citation is Gronnier et al 2022 eLife.

Reply: In the revised manuscript, we updated the reference from “ (Yu et al., 2020)” to “(Gronnier et al., 2022)”.

Question 14: Lines 123-124 - this sentence is incomplete.

Reply: Sorry for the incomplete sentence. Now we reworded the sentence to “In a previous investigation, we demonstrated that flg22 induces FLS2 translocation from AtFlot1-negative to AtFlot1-positive nanodomains in the plasma membrane, implying a connection between FLS2 phosphorylation and membrane nanodomain distribution (Cui et al., 2018). To validate this, we assessed the association of FLS2/FLS2S938D/FLS2S938A with membrane microdomains, using AtRem1.3-associated microdomains as representatives (Huang et al., 2019).” in the revised manuscript.

Question 15: Line 126 - this requires a reference.

Reply: Yes, we added a new reference: “(Huang et al., 2019)” in the revised manuscript.

Question 16: Lines 125-128 - should clarify that the authors are not looking at direct interaction between FLS2 and REM1.3.

Reply: Sorry for the inappropriate expressions here. In the revised manuscript, we reworded the sentence as follows: “To validate this, we assessed the association of FLS2/FLS2S938D/FLS2S938A with membrane microdomains, using AtRem1.3-associated microdomains as representatives (Huang et al., 2019)” .

Question 17: Line 138 - these are odd references to use for such a broad statement.

Reply: Now the inappropriate references cited here have been deleted.

Question 18: Line 161 - incorrect reference, again.

Reply: Sorry for this mistake. In the revised manuscript, we reworded the sentence and changed the reference.

Question 19: Lines 160-165 - this is very confusing and misleading. I would suggest just having a short section introducing PTI earlier on (with appropriate references).

Reply: As suggestion, we reworded and added a section in the revised manuscript as follows: “PTI plays a pivotal role in host defense against pathogenic infections (Lorrai et al., 2021; Ma et al., 2022). Previous studies demonstrated that FLS2 perception of flg22 initiates a complex signaling network with multiple parallel branches, including calcium burst, mitogen-activated protein kinases (MAPKs) activation, callose deposition, and seedling growth inhibition (Baral et al., 2015; Marcec et al., 2021; Huang et al., 2023). Our focus was to investigate the significance of Ser-938 phosphorylation in flg22-induced plant immunity. Figure 4A-F illustrates diverse immune responses in FLS2 and FLS2S938D plants following flg22 treatment. These responses encompass calcium burst activation, MAPKs cascade reaction, callose deposition, hypocotyl growth inhibition, and activation of immune-responsive genes. In contrast, FLS2S938A (Figure S4A-D) exhibited limited immune responses, underscoring the importance of Ser-938 phosphorylation for FLS2-mediated PTI responses”.

Question 20: Line 166 - these are not appropriate references, again.

Reply: Thank you for the suggestion. In the revised manuscript, we removed the inappropriate references. Besides, we added new references by citing: “(Baral et al., 2015; Marcec et al., 2021)”.

Question 21: Lines 169-173 - this is not relevant, the inhibition of growth by elicitors is extremely well-documented (though not by the refs cited here).

Reply: We reworded the sentence and deleted the inappropriate reference in the revised manuscript.

Question 22: Lines 174-175 - I don't see why this is unexpected, as nanodomain organization of PRRs has been previously described.

Reply: Sorry for the inappropriate expressions here. As we have dramatically changed the manuscript, this sentence was deleted from the new version.

References we added into the revised manuscript

Baral A, Irani NG, Fujimoto M, Nakano A, Mayor S, Mathew MK. 2015. Salt-induced remodeling of spatially restricted clathrin-independent endocytic pathways in Arabidopsis root. Plant Cell 27:1297-315. DOI: 10.1105/tpc.15.00154, PMID: 25901088

Barbieri L, Colin-York H, Korobchevskaya K, Li D, Wolfson DL, Karedla N, Schneider F, Ahluwalia BS, Seternes T, Dalmo RA, Dustin ML, Li D, Fritzsche M. 2021. Two-dimensional TIRF-SIM-traction force microscopy (2D TIRF-SIM-TFM). Nature Communications 12:2169. DOI: 10.1038/s41467-021-22377-9, PMID: 33846317

Bertot L, Grassart A, Lagache T, Nardi G, Basquin C, Olivo-Marin J, Sauvonnet N. 2018. Quantitative and statistical study of the dynamics of clathrin-dependent and -independent endocytosis reveal a differential role of endophilinA2. Cell Reports 22: 1574–1588. DOI:org/10.1016/j.celrep.2018.01.039, PMID: 29425511

Bücherl CA, Jarsch IK, Schudoma C, Segonzac C, Mbengue M, Robatzek S, MacLean D, Ott T, Zipfel C. 2017. Plant immune and growth receptors share common signalling components but localise to distinct plasma membrane nanodomains. eLife 6:e25114. DOI: https://doi.org/10.7554/eLife.25114, PMID: 28262094

Chen Y, Munteanu AC, Huang YF, Phillips J, Zhu Z, Mavros M, Tan W. 2009. Mapping receptor density on live cells by using fluorescence correlation spectroscopy. Chemistry 15:5327-36. DOI: https://doi.org/10.1002/chem.200802305, PMID: 19360825

Chinchilla, D., Bauer, Z., Regenass, M., Boller, T., and Felix, G. 2006. The Arabidopsis receptor kinase FLS2 binds flg22 and determines the specificity of flagellin perception. Plant Cell 18:465-476. doi:10.1105/tpc.105.036574, PMID: 16377758

Gada KD, Kawano T, Plant LD, Logothetis DE. 2022. An optogenetic tool to recruit individual PKC isozymes to the cell surface and promote specific phosphorylation of membrane proteins. The Journal of Biological Chemistry 298:101893. DOI: https://doi.org/10.1016/j.jbc.2022.101893, PMID: 35367414

Gronnier J, Franck CM, Stegmann M, DeFalco TA, Abarca A, von Arx M, Dünser K, Lin W, Yang Z, Kleine-Vehn J, Ringli C, Zipfel C. 2022. Regulation of immune receptor kinase plasma membrane nanoscale organization by a plant peptide hormone and its receptors. eLife 11:e74162. DOI: https://doi.org/10.7554/eLife.74162, PMID: 34989334

Hohmann U, Lau K, Hothorn M. 2017. The structural basis of ligand perception and signal activation by receptor kinases. Annual Review of Plant Biology 68:109–137. DOI: https://doi.org/10.1146/annurev-arplant-042916-040957, PMID: 28125280.

Huang D, Sun Y, Ma Z, Ke M, Cui Y, Chen Z, Chen C, Ji C, Tran TM, Yang L, Lam SM, Han Y, Shu G, Friml J, Miao Y, Jiang L, Chen X. 2019. Salicylic acid-mediated plasmodesmal closure via Remorin-dependent lipid organization. Proceedings of the National Academy of Sciences 116:21274–21284. DOI: https://doi.org/10.1073/pnas.1911892116, PMID: 31575745

Huang Y, Cui J, Li M, Yang R, Hu Y, Yu X, Chen Y, Wu Q, Yao H, Yu G, Guo J, Zhang H, Wu S, Cai Y. 2023. Conservation and divergence of flg22, pep1 and nlp20 in activation of immune response and inhibition of root development. Plant Science 331:111686. DOI: https://doi.org/10.1016/j.plantsci.2023.111686, PMID: 36963637

Jiao C, Gong J, Guo Z, Li S, Zuo Y, Shen Y. 2022. Linalool activates oxidative and calciμm burst and CAM3-ACA8 participates in calciμm recovery in Arabidopsis leaves. International Journal of Molecular Sciences, 23:5357. DOI: https://doi.org/10.3390/ijms23105357, PMID: 35628166

Kim TJ, Lei L, Seong J, Suh JS, Jang YK, Jung SH, Sun J, Kim DH, Wang Y. 2018. Matrix rigidity-dependent regulation of Ca2+ at plasma membrane microdomains by FAK visualized by fluorescence resonance energy transfer. Advanced science, 6:1801290. DOI: https://doi.org/10.1002/advs.201801290, PMID: 30828523

Kontaxi C, Kim N, Cousin MA. 2023. The phospho-regulated amphiphysin/endophilin interaction is required for synaptic vesicle endocytosis. Journal of Neurochemistry 166:248–264. DOI: https://doi.org/10.1111/jnc.15848, PMID: 37243578

Lee Y, Phelps C, Huang T, Mostofian B, Wu L, Zhang Y, Tao K, Chang YH, Stork PJ, Gray JW, Zuckerman DM, Nan X. 2019. High-throughput, single-particle tracking reveals nested membrane domains that dictate KRasG12D diffusion and trafficking. eLife 8:e46393. DOI: https://doi.org/10.7554/eLife.46393, PMID: 31674905

Li B, Meng X, Shan L, He P. 2016a. Transcriptional regulation of pattern-triggered immunity in plants. Cell Host Microbe 19:641-50. DOI: 10.1016/j.chom.2016.04.011, PMID: 27173932

Li L, Kim P, Yu L, Cai G, Chen S, Alfano JR, Zhou JM. 2016b. Activation-dependent destruction of a co-receptor by a pseudomonas syringae effector dampens plant immunity. Cell Host Microbe 20:504-514. DOI: https://doi.org/10.1016/j.chom.2016.09.007, PMID: 27736646.b

Lorrai R, Ferrari S. 2021. Host cell wall damage during pathogen infection: mechanisms of perception and role in plant-pathogen interactions. Plants (Basel) 10:399. DOI: https://doi.org/10.3390/plants10020399, PMID: 33669710

Marcec MJ, Tanaka K. 2021. Crosstalk between Calcium and ROS signaling during flg22-triggered immune response in Arabidopsis leaves. Plants 11:14. DOI: 10.3390/plants11010014. PMID: 35009017

Ma M, Wang W, Fei Y, Cheng HY, Song B, Zhou Z, Zhao Y, Zhang X, Li L, Chen S, Wang J, Liang X, Zhou JM. A surface-receptor-coupled G protein regulates plant immunity through nuclear protein kinases. 2022. Cell Host Microbe 30:1602-1614. DOI: 10.1016/j.chom.2022.09.012. Epub 2022 Oct 13. PMID: 36240763.

Martinière A, Zelazny E. 2021. Membrane nanodomains and transport functions in plant. Plant Physiology 187:1839–1855. DOI: https://doi.org/10.1093/plphys/kiab312, PMID: 35235669

Majhi, B.B., Sobol, G., Gachie, S., Sreeramulu, S., and Sessa, G. 2021. BRASSINOSTEROID-SIGNALLING KINASES 7 and 8 associate with the FLS2 immune receptor and are required for flg22-induced PTI responses. Molecular Plant Pathology 22:786-799. DOI:https://doi.org/10.1111/mpp.13062, PMID: 33955635

Mitra SK, Chen R, Dhandaydham M, Wang X, Blackburn RK, Kota U, Goshe MB, Schwartz D, Huber SC, Clouse SD. 2015. An autophosphorylation site database for leucine-rich repeat receptor-like kinases in *Arabidopsis thaliana*. The Plant Journal 82:1042–1060. DOI: https://doi.org/10.1111/tpj.12863, PMID: 25912465

Orosa B, Yates G, Verma V, Srivastava AK, Srivastava M, Campanaro A, De Vega D, Fernandes A, Zhang C, Lee J, Bennett MJ, Sadanandom A. 2018. SμmO conjugation to the pattern recognition receptor FLS2 triggers intracellular signalling in plant innate immunity. Nature Communications 9:5185. DOI: https://doi.org/10.1038/s41467-018-07696-8, PMID: 30518761

Sun Y, Li L, Macho AP, Han Z, Hu Z, Zipfel C, Zhou JM, Chai J. 2013. Structural basis for flg22-induced activation of the Arabidopsis FLS2-BAK1 immune complex. Science 342:624-628. DOI: https://doi.org/10.1126/science.1243825, PMID: 24114786

Vitrac H, Mallampalli VKPS, Dowhan W. 2019. Importance of phosphorylation/dephosphorylation cycles on lipid-dependent modulation of membrane protein topology by posttranslational phosphorylation. The Journal of Biological Chemistry 294:18853–18862. DOI: https://doi.org/10.1074/jbc.RA119.010785, PMID: 31645436

Xue Y, Xing J, Wan Y, Lv X, Fan L, Zhang Y, Song K, Wang L, Wang X, Deng X, Baluška F, Christie JM, Lin J. 2018. Arabidopsis blue light receptor phototropin 1 undergoes blue light-induced activation in membrane microdomains. Molecular Plant 11:846-859. DOI: 10.1016/j.molp.2018.04.003, PMID: 29689384

Xing J, Ji D, Duan Z, Chen T, Luo X. 2022. Spatiotemporal dynamics of FERONIA reveal alternative endocytic pathways in response to flg22 elicitor stimuli. New Phytologist 235: 518-532. DOI: 10.1111/nph.18127, PMID: 35358335

Zhai K, Liang D, Li H, Jiao F, Yan B, Liu J, Lei Z, Huang L, Gong X, Wang X, Miao J, Wang Y, Liu JY, Zhang L, Wang E, Deng Y, Wen CK, Guo H, Han B, He Z. 2021. NLRs guard metabolism to coordinate pattern- and effector-triggered immunity. Nature 601:245-251. DOI: https://doi.org/10.1038/s41586-021-04219-2, PMID: 34912119

Zhong YH, Guo ZJ, Wei MY, Wang JC, Song SW, Chi BJ, Zhang YC, Liu JW, Li J, Zhu XY, Tang HC, Song LY, Xu CQ, Zheng HL. 2023. Hydrogen sulfide upregulates the alternative respiratory pathway in mangrove plant Avicennia marina to attenuate waterlogging-induced oxidative stress and mitochondrial damage in a calciμm-dependent manner. Plant Cell and Environment 46:1521-1539. DOI: https://doi.org/10.1111/pce.14546, PMID: 36658747

Inappropriate references we deleted from the revised manuscript

Schulze S, Yu L, Hua C, Zhang L, Kolb D, Weber H, Ehinger A, Saile SC, Stahl M, Franz-Wachtel M, Li L, El Kasmi F, Nürnberger T, Cevik V, Kemmerling B. 2022. The Arabidopsis TIR-NBS-LRR protein CSA1 guards BAK1-BIR3 homeostasis and mediates convergence of pattern- and effector-induced immune responses. Cell Host Microbe 30:1717-1731.e6. DOI: 10.1016/j.chom.2022.11.001, PMID: 36446350

Wang Q, Zhao Y, Luo W, Li R, He Q, Fang X, Michele RD, Ast C, von Wirén N, Lin J. 2013. Single-particle analysis reveals shutoff control of the Arabidopsis ammonium transporter AMT1;3 by clustering and internalization. Proceedings of the National Academy of Sciences of the United States of America 110:13204-9. DOI: 10.1073/pnas.1301160110, PMID: 23882074

Eichel K, Jullié D, von Zastrow M. β-Arrestin drives MAP kinase signalling from clathrin-coated structures after GPCR dissociation. Nature Cell Biology 18:303-10. DOI: 10.1038/ncb3307, PMID: 26829388

Van Itallie CM, Anderson JM. Phosphorylation of tight junction transmembrane proteins: Many sites, much to do. Tissue Barriers 6:e1382671. DOI: 10.1080/21688370.2017.1382671, PMID: 29083946

Monje-Galvan V, Warburton L, Klauda JB. Setting up all-atom molecular dynamics simulations to study the interactions of peripheral membrane proteins with model lipid bilayers. Methods in Molecular Biology 1949:325-339. DOI: 10.1007/978-1-4939-9136-5_22, PMID: 30790265.

Trotta A, Bajwa AA, Mancini I, Paakkarinen V, Pribil M, Aro EM. The role of phosphorylation dynamics of CURVATURE THYLAKOID 1B in plant thylakoid membranes. Plant Physiology 181:1615-1631. DOI: 10.1104/pp.19.00942, PMID: 31615849

Dorrity MW, Saunders LM, Queitsch C, Fields S, Trapnell C. Dimensionality reduction by UMAP to visualize physical and genetic interactions. Nature Communications 11:1537. DOI: 10.1038/s41467-020-15351-4, PMID: 32210240

Sato KI, Tokmakov AA. Membrane microdomains as platform to study membrane-associated events during Oogenesis, Meiotic Maturation, and Fertilization in *Xenopus laevis*. Methods in Molecular Biology 920:59-73. DOI: 10.1007/978-1-4939-9009-2_5, PMID: 30737686.

Ozolina NV, Kapustina IS, Gurina VV, Bobkova VA, Nurminsky VN. Role of plasmalemma microdomains (Rafts) in protection of the plant cell under Osmotic stress. Journal of Membrane Biology 254:429-439. DOI: 10.1007/s00232-021-00194-x, PMID: 34302495

Boutté Y, Moreau P. Plasma membrane partitioning: from macro-domains to new views on plasmodesmata. Frontiers in Plant Science 5:128. DOI: 10.3389/fpls.2014.00128. PMID: 24772114

Yu M, Cui Y, Zhang X, Li R, Lin J. Organization and dynamics of functional plant membrane microdomains. Cellular and Molecular Life Sciences 77:275-287. DOI: 10.1007/s00018-019-03270-7, PMID: 31422442

Zhao Z, Li M, Zhang H, Yu Y, Ma L, Wang W, Fan Y, Huang N, Wang X, Liu K, Dong S, Tang H, Wang J, Zhang H, Bao Y. Comparative proteomic analysis of plasma membrane proteins in rice leaves reveals a vesicle trafficking network in plant immunity that is provoked by Blast Fungi. Frontiers in Plant Science 13:853195. DOI: 10.3389/fpls.2022.853195, PMID: 35548300

Hilgemann DW, Dai G, Collins A, Lariccia V, Magi S, Deisl C, Fine M. Lipid signaling to membrane proteins: From second messengers to membrane domains and adapter-free endocytosis. Journal of General Physiology 150:211-224. DOI: 10.1085/jgp.201711875, PMID: 29326133

Joshi R, Paul M, Kumar A, Pandey D. Role of calreticulin in biotic and abiotic stress signalling and tolerance mechanisms in plants. Gene 714:144004. DOI: 10.1016/j.gene.2019.144004, PMID: 31351124

Chen Y, Cao C, Guo Z, Zhang Q, Li S, Zhang X, Gong J, Shen Y. Herbivore exposure alters ion fluxes and improves salt tolerance in a desert shrub. Plant Cell and Environment 43:400-419. DOI: 10.1111/pce.13662, PMID: 31674033

Chi Y, Wang C, Wang M, Wan D, Huang F, Jiang Z, Crawford BM, Vo-Dinh T, Yuan F, Wu F, Pei ZM. Flg22-induced Ca2+ increases undergo desensitization and resensitization. Plant Cell and Environment 44:3563-3575. DOI: 10.1111/pce.14186, PMID: 34536020

Zhang M, Su J, Zhang Y, Xu J, Zhang S. Conveying endogenous and exogenous signals: MAPK cascades in plant growth and defense. Current Opinion in Plant Biology 45:1-10. DOI: 10.1016/j.pbi.2018.04.012, PMID: 29753266

Arnaud D, Deeks MJ, Smirnoff N. RBOHF activates stomatal immunity by modulating both reactive oxygen species and apoplastic pH dynamics in Arabidopsis. Plant Journal 116:404-415. DOI: 10.1111/tpj.16380, PMID: 37421599

Zou Y, Wang S, Zhou Y, Bai J, Huang G, Liu X, Zhang Y, Tang D, Lu D. Transcriptional regulation of the immune receptor FLS2 controls the ontogeny of plant innate immunity. Plant Cell.30:2779-2794. DOI: 10.1105/tpc.18.00297, PMID: 30337428

Ngou BPM, Jones JDG, Ding P. Plant immune networks. Trends in Plant Science 27:255-273. DOI: 10.1016/j.tplants.2021.08.012, PMID: 34548213.

Yu M, Liu H, Dong Z, Xiao J, Su B, Fan L, Komis G, Šamaj J, Lin J, Li R. 2017. The dynamics and endocytosis of Flot1 protein in response to flg22 in Arabidopsis. Journal of Plant Physiology 215:73–84. DOI: https://doi.org/10.1016/j.jplph.2017.05.010, PMID: 28582732